# Sampling from Your Language Model One Byte at a Time

## Abstract

Tokenization is used almost universally by modern language models, enabling efficient text representation using multi-byte or multi-character tokens. However, prior work has shown that tokenization can introduce distortion into the model's generations, an issue known as the *Prompt Boundary Problem (PBP)*. For example, users are often advised not to end their prompts with a space because it prevents the model from including the space as part of the next token. While this heuristic is effective in English, the underlying PBP continues to affect code generation and languages such as Chinese, where tokens often do not line up with word and syntactic boundaries. In this work, we present an inference-time method to convert any autoregressive LM with a BPE tokenizer into a character-level or byte-level LM. Our method efficiently solves the PBP and is also able to unify the vocabularies of language models with different tokenizers, allowing one to ensemble LMs with different tokenizers at inference time or transfer the post-training from one model to another using proxy-tuning. We demonstrate in experiments that the ensemble and proxy-tuned models outperform their constituents on downstream evals.

## 1 Introduction

Tokenization is a crucial component of nearly all modern language models: it allows them to efficiently consume and produce arbitrary streams of text using only finite vocabularies. The vast majority of tokenizers in use today, such as those based on Byte-Pair Encoding (BPE) (Sennrich et al., 2016) or Unigram (Kudo & Richardson, 2018), feature tokens spanning multiple bytes or characters, allowing them to represent text more efficiently than purely byte-level or character-level tokenization (Clark et al., 2022; Xue et al., 2022; Wang et al., 2024).

Users of LMs are generally unaware of the tokenization and expect LMs to operate on strings over an alphabet $\Sigma$, consuming a $\mathrm{prompt} \in \Sigma^*$ as a string and producing a string $\mathrm{completion} \in \Sigma^*$ thereof. Tokenized LMs emulate this by *(i)* encoding the text into a sequence of tokens, *(ii)* sampling a completion of the token sequence with the LM, and *(iii)* decoding the generated token sequence back into text. This corresponds to the distribution $\mathrm{completion} \sim \mathrm{decode}(t_{k+1}, \ldots, t_n)$ where

$$\mathsf{P}(t_{k+1}, \ldots, t_n \mid (t_1, \ldots, t_k) = \mathrm{encode}(\mathrm{prompt})) \;=\; \prod_{i=k+1}^{n} \mathsf{P}(t_i \mid t_1, \ldots, t_{i-1}) \,, \qquad (1)$$

P represents the token-level distribution of the model and $\mathrm{encode} \colon \Sigma^* \to V^*$ and $\mathrm{decode} \colon V^* \to \Sigma^*$ translate between strings and token sequences over vocabulary $V$.

While sampling from this distribution is easy, it can also produce undesirable results. This is sometimes called the *prompt boundary problem* (Lundberg, 2023; Vieira et al., 2024; Phan et al., 2024), which we describe below.

**The Prompt Boundary Problem (PBP).** In particular, Eq. (1) introduces distortion whenever the prompt ends on a prefix of what could otherwise be a single token. More concretely, consider LLAMA-3.2-1B and suppose the user's prompt ends with the text "becau" (`["bec"` = `17106, "au"` = `2933]` as tokens): The user most likely expects the continuation to begin with "se" (325) since "because" is a common word. However during training, the model has only ever seen the word "because" represented as a single token (11458) and never as the sequence [17106, 2933, 325]. Accordingly, the actual next token LLAMA-3.2-1B predicts

is "z" (89) which, while plausible in some scenarios, is an arguably unlikely continuation representing an artifact of tokenization. While this example may seem contrived at first glance, there are many situations where this problem may arise (Fig. 1 shows a few more examples):

1. In languages that do not separate words with whitespace, such as Chinese and Japanese, tokens can span multiple words, so this issue can arise even when the prompt ends with a complete word.

2. Any tokenizer that features multi-word tokens, which can bring gains in encoding efficiency (Gee et al., 2023; Kumar & Thawani, 2022; Liu et al., 2025; Tănase & Pelican, 2025), suffer from the same problem as Chinese and Japanese.

3. When completing code, it is common to request completions while in the middle of an identifier (Jackson, 2025; Bavarian et al., 2022).

```
> olmo.generate(tok.encode("This a tes"))
"erstor" ✘
> ByteSampler(olmo, "This is a tes")
"t" ✔
                        Japan's capital is Tokyo    China's capital
> qwen.generate(tok.encode("日本的首都是东京，中国的首都"))
    also is Beijing
"也是北京" ✘
> ByteSampler(qwen, "日本的首都是东京，中国的首都")
    is Beijing
"是北京" ✔
> olmo.generate(tok.encode("document.getElement"))
"('div')" ✘
> ByteSampler(olmo, "document.getElement")
"ById('button')" ✔
```

Figure 1: ByteSampler resolves the prompt boundary problem (exhibited in the output of generate()). In this example, ␣test, 都是, and .getElementById are all single tokens in the respective tokenizers.

4. This issue also occurs when performing constrained generation from language models (Ribeiro, 2023; Beurer-Kellner et al., 2024).

In general, the user, unaware of the tokenization, expects samples from the distribution conditioned on the byte-string prefix prompt,

$$P(t_1, \ldots, t_n \mid \text{prompt} \sqsubseteq \text{decode}(t_1, \ldots, t_n)) , \tag{2}$$

where $\sqsubseteq$ denotes the sequence prefix relation. As we just saw, the token-prefix conditioned distribution of Eq. (1) and the *byte-prefix conditioned distribution* of Eq. (2) can differ substantially (e.g., Fig. 1). Eq. (2) transcends the arbitrary token boundary set where the user provided prompt stops, decoupling the prompt boundary from token boundaries, allowing the language model to choose the most natural segmentation of the prompt. This leads to a fundamental algorithmic question of interest: how do we efficiently sample from the byte-prefix conditioned distribution of Eq. (2)?

**Contributions.** We introduce an efficient procedure to condition a BPE tokenizer-based model on an arbitrary *byte-prefix* given only access to the tokenizer and log-probability queries to the model (Section 3). We demonstrate in experiments that this represents an exact solution to the Prompt Boundary Problem presented above (Section 4.2). We show that our method can be used to convert the model into a byte-level language model and that this ability can be used to unify the vocabularies of different models. This enables exact byte-level ensembles of language models with different tokenizers (Section 4.3) and allows one to transfer the post-training of one model onto another model at inference time using proxy-tuning (Liu et al., 2024c) (Section 4.4). We demonstrate in proof-of-concept experiments that language model ensembles and proxy-tuned models constructed with our method are able to outperform their constituent models in downstream evaluations.

## 2 BACKGROUND

In this section we give essential background regarding tokenization as well a prior work addressing the Prompt Boundary Problem. We discuss additional related works in Appendix A.

**Byte Pair Encoding**. BPE was originally presented as a form of data compression in Gage (1994) and was proposed for use in NLP in Sennrich et al. (2016). To tokenize a piece of text with a typical BPE-based tokenizer, the text is first split into chunks, a process called *pretokenization*. These chunks, or *pretokens*, are then tokenized separately using BPE (thus no token may cross the boundary between pretokens). The BPE tokenizer processes each pretoken by first converting the text into a sequence of elements of the tokenizer's base vocabulary (common choices for base vocabulary are individual

| | Tokenizers | Exact | Overhead (CPU) | Inference tokens (GPU) |
|---|---|---|---|---|
| Backtracking | Any | No | $O(1)$ | $O(1)$ |
| Prefix Covering (Vieira et al., 2024) | Any | Yes | $2^{O(n)}$ | $2^{O(n)}$ |
| Back Tokenization (Turaga, 2025) | S | Yes | $O(n)$ | $O(1)$ (optimal) |
| (Phan et al., 2025) | S | Yes | $O(n)$ | $O(1)$ |
| ByteSampler (ours) | H | Yes | $O(1)$ | $O(1)$ (optimal) |

Table 1: **Comparison of various mitigations for the prompt boundary problem:** we list tokenizers supported (S for SentencePiece BPE and H for HuggingFace ByteLevel BPE, see Appendix C.6 for details and exceptions) and complexity (in both CPU overhead and inference tokens) when sampling each new character while generating an $n$ character string. Our method has the same complexity as backtracking methods (Ribeiro, 2023; Dagan et al., 2024; Athiwaratkun et al., 2024) while remaining exact, i.e., matching Eq. (2) in distribution, modulo invalid sequences (see below for discussion). We report the LM inference complexity upper bounds using analysis from Section 3.1 when using prefix caching. "(optimal)" indicates that the token evaluations for any input will be the minimum required for exactness.

characters or bytes under UTF-8 encoding). Next, an ordered list of merges is applied to the sequence to form larger tokens. Each merge specifies a contiguous pair of tokens (which may include products of previous merges), and a new token that represents their concatenation. The merges are applied left-to-right and once all valid merges are applied, the tokenization is complete.

**Prompt Boundary Problem**. Issues surrounding tokenization have been extensively documented in prior work. The prompt boundary problem was presented for maximum prefix encoding in Phan et al. (2024) and for BPE tokenizers in Vieira et al. (2024) and Ribeiro (2023). Many methods have been proposed to address the prompt boundary issue. One line of heuristic techniques, including token healing (Ribeiro, 2023) and its generalizations (Dagan et al., 2024; Athiwaratkun et al., 2024) perform "backtracking" by *(i)* removing one or more of the most recent tokens, followed by *(ii)* sampling a continuation of the partial prompt using the language model, constraining the newly generated tokens to match the remaining prompt.

Exact methods, which preserve the sampling distribution of the original language model as shown in (5), have also been proposed. Vieira et al. (2024) gave an exact method which requires exponential time as well as an approximate solution leveraging beam search. Turaga (2025) proposed a method that combines backtracking with the exponential time method of Vieira et al. (2024), adding a "back tokenization" step that significantly reduces the number of necessary calls to the language model, but still requires exponential overhead. Additionally, Phan et al. (2024; 2025) proposed an exact method which requires only linear time.

Although all of the above methods, except for Backtracking, are "exact," they may produce slightly different sampling distributions. This is because the methods differ in their handling of invalid token sequences, which are sequences that can never be output by the tokenizer, but can still be generated erroneously by the model. For now, we will assume that the model will always produce valid token sequences, in which case all of the exact methods are identical. We discuss this assumption and the differences when it does not hold in more detail in Appendix D.

## 3 METHOD

In this section, we present the building blocks which we use to construct a procedure for sampling from a tokenizer-based language model one byte at a time. The fundamental structure of the algorithm is based on what we call the Valid Covering Tree, which is the tree of all possible valid token sequences that share a specific byte prefix and do not extend past the end of the prefix by more than one full token. We show the construction of the Valid Covering Tree in Fig. 2.

The tree depicted in Fig. 2b corresponds to the *cover* described in Vieira et al. (2024), which notes that it will generally have exponential size in the length of the prefix. In contrast, the Valid Covering Tree, which is a subtree of the one in Fig. 2b, has several properties which will prove useful:

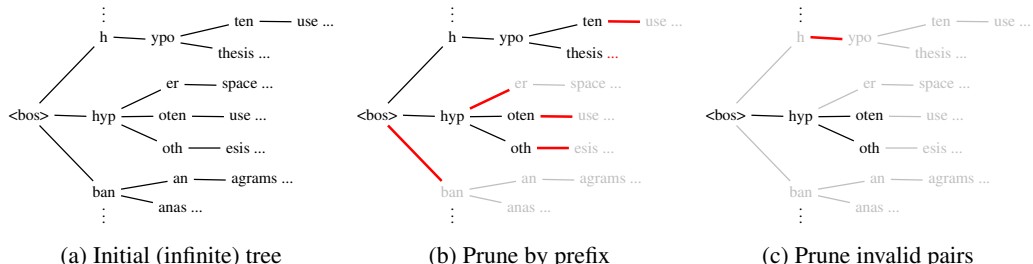

|  (a) Initial (infinite) tree  |  (b) Prune by prefix  |  (c) Prune invalid pairs  |

Figure 2: *Construction of the Valid Covering Tree* for string prefix "hypot": (a) starting with the infinite tree of all possible token sequences (many edges not shown), we prune branches that (b) do not match the given prefix or begin after the prefix ends or (c) contain invalid contiguous pairs of tokens. More example trees are shown in Appendix G.

1. **Correctness:** The tree represents exactly the set of valid sequences of tokens with the prompt as a prefix. (See Section 3.1 and Appendix D.)

2. **Compactness:** The tree is composed of a "trunk" of tokens that are fully determined (starting at the root, every node has only one child) plus a finite number of "branching" nodes at the end of the trunk. (The number is bounded by a constant which depends only on the tokenizer, see proof in Appendix D.1.)

Additional implementation details and optimizations are presented in Appendix C.

### 3.1 PAIRWISE VALIDATION

Recall that a token sequence is *valid* if it is the encoding of some string under the BPE encoder.[1] The correctness of the pairwise pruning depends on the following proposition regarding validity under BPE tokenization. (We give a full proof in Appendix D.2.)

**Proposition 3.1.** *Let* (encode, decode) *denote a BPE encoder and decoder pair corresponding to some merge list $M$ and vocabulary $V$. We call a token sequence $T = [t_1, t_2, \ldots, t_n] \in V^n$ valid if* $\mathrm{encode}(\mathrm{decode}(T)) = T$. *Then $T$ is valid if and only if $[t_i, t_{i+1}]$ is valid for all $i \in \{0, \ldots, n-1\}$.*

To see that this proposition is true, consider two valid token sequences $T_1 = \mathrm{encode}(S_1)$ and $T_2 = \mathrm{encode}(S_2)$ and note that the concatenation $T_1 + T_2$ is valid if and only if there is no merge applied that crosses the boundary between $S_1$ and $S_2$ while tokenizing $S_1 + S_2$. We depict an example of both cases using OpenAI's cl100k tokenizer (OpenAI, 2023) in Fig. 3.[2]

If there is no such merge, then the two strings are effectively tokenized separately, so $\mathrm{encode}(S_1 + S_2) = T_1 + T_2$ and thus $T_1 + T_2$ is valid. On the other hand, if there is such a merge, then $\mathrm{encode}(S_1 + S_2)$ must feature a token crossing the boundary (since no merge can be undone), which means $T_1 + T_2$ cannot be valid since it has no such token.

This implies a fast method to test whether a pair of tokens is valid: we inspect the merge trajectory along the boundary between the tokens and check if any conflicting merges would be applied. The worst case merge tree depth is fixed by the tokenizer, so this check can be done in constant time.[3]

### 3.2 LANGUAGE MODELING USING VALID COVERING TREES

Now that we have defined the Valid Covering Tree, we can use it to perform various common language modeling operations. Given a byte-string $S$, we compute its VCT $T$:

---

[1]The notion of pairwise validation of token sequences was first used in van Antwerpen & Neubeck (2025) as the basis for a streaming algorithm and a fast backtracking-based algorithm for BPE tokenization (without addressing the PBP).

[2]It's worth noting that the analogs of Proposition 3.1 do *not* hold for either Unigram (Kudo & Richardson, 2018) or Wordpiece (Schuster & Nakajima, 2012) tokenizers.

[3]We generally expect the depth of the merge trees to scale with the logarithm of the vocabulary size $V$, although we ignore scaling with respect to the tokenizer's parameters for brevity.

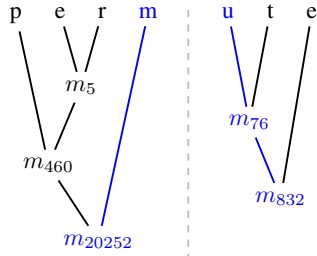
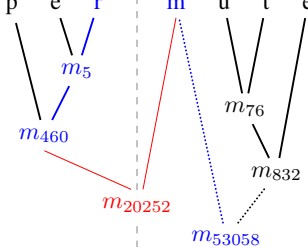

(a) Valid pair: no merge crossing boundary    (b) Invalid pair: merge $m_{20252}$ crosses boundary

Figure 3: Example of valid and invalid token pairs. We show the initial string's bytes and the merges $m_t \in M$ that are applied to the string (in order of $t$) to tokenize the string. In the invalid case, merge $m_{53058}$ cannot occur because a conflicting merge $m_{20252}$ was applied earlier. The key observation is that we only need to consider the trajectory at the boundary (in blue) to decide if the pair is valid.

1. To **compute the probability of** $S$ **(as a prefix)** under the LM, we sum the cumulative probabilities the LM assigns to the sequences represented by all leaves of $T$.

2. To **sample a completion of** $S$ **while avoiding the PBP**, we compute the probability (as above) of every leaf in $T$ and sample one of them accordingly. We are then free to continue sampling a continuation from that leaf using normal token-level sampling. This can be used to solve the PBP without paying the cost of byte-level sampling.

3. To **compute the next byte distribution following** $S$, we group the leaves of $T$ by their corresponding next byte and sum the probabilities of the leaves in each group. This can be combined with a sampling rule to generate text one byte at a time. Naturally, this will generate text more slowly than sampling at the token level. We quantify this overhead in Section 4.2.

We use "ByteSampler" to refer to this collection of capabilities for convenience.

### 3.3 Incrementally Updating the Valid Covering Tree

We can use the Valid Covering Tree as the basis for a streaming algorithm by incrementally updating it to reflect newly sampled bytes. Given a stream of input bytes, we will use Algorithm 1 to update "branches" of the Valid Covering Tree, while writing the fully determined "trunk" of tokens to an output stream.

---

**Algorithm 1:** Streaming BPE tokenization maintaining a tree matching Fig. 2c

---

**Input:** Branching tree $T$, new byte $b$
**Output:** stream of fully determined tokens
**for** *every node $N$ that ends before $b$ begins* **do**
   |   add all *valid* next tokens as children of $N$;           // See Fig. 2c
**end**
Prune branches that do not match $b$;           // See Fig. 2b
**while** *the root of $T$ has only one child* **do**
   |   Add the root token to the output stream and make its only child the new root;
**end**

---

This routine is efficient because the tree $T$ always has a bounded size (as shown in Appendix D.1). This means that both the expansion of nodes to extend the tree, as well as pruning nodes not matching the new byte can be done in constant time. For more concrete performance numbers see Section 4.1, where we show that the tree has only 0.72 extra non-leaf nodes on average.

## 4 Experiments

In our experiments, we apply ByteSampler at inference time to off-the-shelf language models. In Section 4.1 we show that our method has less computational overhead compared to other exact

methods. Next, in Section 4.2, we show that exact methods perform better than heuristics in character-level language modeling. Finally, we present several applications of our method to enable higher-level functions such as ensembling (Section 4.3) and proxy-tuning (Section 4.4) models with mismatched tokenizers.

## 4.1 EFFICIENCY

As discussed in Section 2, there are several existing methods which are also "exact." Although each technically corresponds to a different sampling distribution, we do not expect there to be any significant differences between them in practice. Therefore, the main distinguishing factor to consider is the method's computational cost. To estimate the cost in a realistic setting, we sample a random 100 character substring from the OLMO2 pretraining corpus (OLMo et al., 2024) and estimate how many inference tokens (according to the OLMo2 tokenizer) each method requires to calculate the probability of the substring as a text prefix. Note that the substring is sampled uniformly, so it is about 80% likely to end in the middle of a word. We report the average inference cost in tokens, averaged over 10,000 samples, for several methods in Table 2. We also perform a more detailed comparison with the method of Phan et al. (2025) in Appendix F.1.

| Method | Inference Tokens | Overhead vs. BPE |
|---|---|---|
| No mitigation (plain BPE) | 23.51 | 0 |
| Prefix Covering (Vieira et al., 2024) | $2.12 \times 10^{30}$ | $+2.12 \times 10^{30}$ |
| Phan et al. (2025) | 72.99 | +49.47 |
| Phan et al. (2025) with prefix caching | 25.61 | +2.09 |
| ByteSampler (ours)[4] | **24.24** | **+0.72** |

Table 2: **Inference cost of various exact solutions to the prompt boundary problem.** Our method has 65% less overhead than the next best method. Overhead vs. BPE measures the average additional tokens of inference required by the method, compared to plain BPE. Importantly, the overhead is paid for each byte when sampling at the byte level, making low overhead crucial for efficient sampling.

## 4.2 CHARACTER-LEVEL LANGUAGE MODELING

| Prediction unit | Method | Loss per unit | Bits per character[5] |
|---|---|---|---|
| Token | Plain BPE | 2.67 | 0.85 |
| Character | No mitigation (plain BPE) | 4.81 | 6.94 |
| Character | ByteSampler (ours) | **0.60** | **0.87** |

Table 3: **Language modeling loss of OLMO2-1B on English text using various methods**. We compare three settings: *(i)* the original token-level cross-entropy loss when predicting the next token; *(ii)* the character-level loss when predicting the next character by directly tokenizing the prompt and calculating the next character distribution; and *(iii)* the character-level loss obtained using ByteSampler to predict the next character. The higher loss per unit for token-level prediction is to be expected, as tokens are harder to predict than bytes. Once the loss is normalized to bits per character, our method and the original model achieve similar results, which demonstrates that our method does not degrade language modeling quality.

In this section, we will focus on converting off-the-shelf language models into character-level language models.[6] We then evaluate the character-level prediction performance using the standard cross-entropy loss as well as next-character prediction accuracy in two languages: English in Section 4.2.1 and Chinese in Section 4.2.2.

---

[4]We believe Back Tokenization (Turaga, 2025) should match our method when it comes to required inference tokens. However, its worst-case exponential *overhead* limits its practicality.

[5]For token level prediction, calculated using a conversion rate of 4.518 characters per token.

[6]We choose character-level modeling for this section, even though our method supports byte-level predictions, because some related methods can only operate on character strings.

### 4.2.1 OLMo2 for English Text

In this setting, we sample a document randomly from the OLMO2 pretraining corpus (OLMo et al., 2024) and choose a random prefix of the document of length at most 1000 characters. We then compute the next-character distribution according to OLMO2-1B Team (2025b) using various methods. To allow comparison with the original token-based model, we also truncate the prefix to the nearest token boundary and perform next-token prediction with the original model. We can compare the character-level and token-level losses via bits per character (Mielke, 2019), which normalizes the loss to account for the fact that tokens are more difficult to predict due to their greater information content. We report the average loss of the predictions over 100,000 such documents in Table 3.

From the results in Table 3, we can clearly see the effect of the prompt boundary problem: naively predicting the next character by directly applying the tokenizer to an arbitrary string prefix as in Eq. (1) leads to poor performance ("no mitigation" in Table 3). In contrast, ByteSampler nearly matches the performance of the original token-based model ("plain BPE") in bits per character, as expected for exact methods.

For backtracking methods, it is not easy to compute the probability of any particular next character. This prevents us from calculating the cross-entropy loss as in Table 3. For our experiments, we compare to the Token Alignment method of Athiwaratkun et al. (2024), which is the most advanced of the proposed backtracking methods and also includes token healing as a special case. We use it to directly predict the next character by sampling greedily and report the average accuracy over 100,000 samples in Table 4.

| Method | Next character accuracy | Overhead vs. BPE |
|---|---|---|
| No mitigation (plain BPE) | 29.490 | 0 |
| 1 Token Backtracking (Token Healing) | 71.634 | **+0.43** |
| 2 Token Backtracking (Token Alignment) | 76.281 | +0.53 |
| 4 Token Backtracking (Token Alignment) | 75.407 | +1.08 |
| ByteSampler (ours) | **81.560** | +1.72 |

Table 4: **Next character prediction accuracy of OLMO2-1B on English text using various methods**. We compare three settings *(i)* directly tokenizing the prompt and greedily sampling until the first character of the completion is determined; *(ii)* using backtracking with Token Alignment (of which Token Healing is a special case) to predict the next character; and *(iii)* using ByteSampler to predict the next character. Overhead vs. BPE measures the average additional tokens of inference required by the method, compared to *(i)*.

| Prediction unit | Method | Loss per unit | Bits per character[7] |
|---|---|---|---|
| Token | Plain BPE | 3.43 | 3.50 |
| Character | No mitigation (plain BPE) | 3.79 | 5.47 |
| Character | ByteSampler (ours) | **2.38** | **3.43** |

Table 5: **Language modeling loss of QWEN3-1.7B-BASE on Chinese text using various methods**. We use the same settings and metrics as Table 3. Similarly to our English results, ByteSampler achieves a similar normalized language modeling loss (in bits per character) to the original model which can only perform next token prediction.

Interestingly, we find that too much backtracking hurts the performance of the Token Alignment method. We believe this is because the sampling step often segments the remainder of the prompt in a non-standard way, which may harm the performance of the model.

### 4.2.2 QWEN3 for Chinese Text

Since Chinese writing does not use whitespace, ending the prompt with a complete word does not generally provide a reliable token boundary. This makes it more difficult to heuristically avoid the

---

[7]For token level prediction, calculated using a conversion rate of 1.415 characters per token.

PBP. Similar to Section 4.2.1, we sample a random prefix of length at most 500 characters of a random document from the Chinese subset of the MADLAD-400 dataset (Kudugunta et al., 2023). We then compute the distribution of next characters according to QWEN3-1.7B-BASE (Team, 2025c) using various methods and report the average cross-entropy loss over 100,000 documents in Table 5.

Once again, the naive method fails while our method achieves similar normalized loss to the original token-level model. We also report next character prediction accuracy to allow comparison with backtracking methods. Note that Chinese has much higher entropy at the character level so the average accuracies are proportionally lower.

## 4.3   BYTE-LEVEL ENSEMBLE

Another application enabled by byte-level sampling is the ensembling of language models with different tokenizers. In general, when vocabularies between LMs are the same, their next-token probability or logit distribution can be combined via arithmetic into a single distribution, but this cannot be done directly when the vocabularies differ. Several works have proposed methods to combine LM predictions despite mismatching vocabularies (Kasai et al., 2022; Lv et al., 2024; Liu et al., 2024d; Xu et al., 2024a), but these may introduce bias into the sampling distribution. Our method makes the direct ensemble possible by converting models with BPE tokenizers into a byte-wise models, thus unifying their vocabularies.

In our experiment, we consider an ensemble of three small base language models: QWEN3-1.7B (Team, 2025c), OLMO2-1B OLMo et al. (2024); Team (2025b), and LLAMA3.2-1B Team (2024c). We combine the predictions by computing the average $p_{\text{ensemble}} = \frac{1}{n} \sum_{i=1}^{n} p_i$ where $p_1, \ldots, p_n$ are the next-byte probability distributions for each model. We evaluate the models on a suite of seven tasks and report the results in Table 7.

| Method | Next character accuracy | Overhead vs. BPE |
|---|---|---|
| No mitigation (plain BPE) | 32.8 | 0 |
| 1 Token Backtracking (Token Healing) | 49.2 | +1.82 |
| 2 Token Backtracking (Token Alignment) | 49.6 | +2.98 |
| 4 Token Backtracking (Token Alignment) | 49.0 | +5.30 |
| ByteSampler (ours) | **52.7** | **+1.60** |

Table 6: **Next character prediction accuracy of QWEN3-1.7B-BASE on Chinese text using various methods**. We use the same settings and metrics as Table 4. Similar to our English language results, ByteSampler achieves the best prediction accuracy, but unlike in English, ByteSampler also requires the least overhead of all methods. This highlights that languages with multi-byte characters [8] can behave differently than ones which typically use a single byte for each character.

| Task | QWEN3 | OLMO2 | LLAMA3.2 | Average | Ensemble |
|---|---|---|---|---|---|
| Arithmetic (Brown et al., 2020) | 0.974 | 0.838 | 0.831 | 0.881 | **0.978** |
| DROP (Dua et al., 2019) | 0.470 | 0.409 | 0.299 | 0.393 | **0.479** |
| Jeopardy (Tunguz, 1019) | 0.274 | 0.327 | 0.264 | 0.288 | **0.347** |
| LAMBADA (Paperno et al., 2016) | 0.727 | 0.628 | 0.510 | 0.622 | **0.755** |
| SQuAD (Rajpurkar et al., 2016) | **0.845** | 0.802 | 0.694 | 0.780 | 0.836 |
| TriviaQA (Joshi et al., 2017) | 0.389 | **0.535** | 0.443 | 0.456 | 0.526 |
| WikidataQA (BIG-bench, 2023) | 0.689 | 0.643 | 0.658 | 0.663 | **0.719** |

Table 7: **Byte-level ensemble results.** We report the performance (accuracy) of a byte-level ensemble of three models on downstream evals, along with the individual performance of each model. We see that the ensemble is competitive with the best individual model on each task and consistently outperforms the average performance across the three models. All 95% confidence intervals are smaller than $\pm 0.014$. We give more details regarding the evaluation in Appendix B.2.

---

[8]Chinese typically uses three bytes for each character when encoded using UTF-8.

## 4.4 BYTE-LEVEL PROXY-TUNING

In addition to additive ensembles over probabilities, the *logit*-level predictions of multiple LMs can be combined via arithmetic, with individual LMs acting as "experts" (if their predictions are combined additively) or "anti-experts" (if subtractively) (Liu et al., 2021; Li et al., 2023a; Shi et al., 2024b; Gera et al., 2023; Chuang et al., 2024; Shi et al., 2024a). In particular, this form of ensembling can be used to achieve the *effect* of tuning a large pretrained LM without accessing model weights. To see how this can be done, note that clearly for logit vectors

$$\boldsymbol{\ell}_{\text{tuned}} = \boldsymbol{\ell}_{\text{base}} + (\boldsymbol{\ell}_{\text{tuned}} - \boldsymbol{\ell}_{\text{base}}).$$

The idea of *proxy-tuning* Liu et al. (2024c) is to approximate the term $\boldsymbol{\ell}_{\text{tuned}} - \boldsymbol{\ell}_{\text{base}}$ using the difference between a pair of tuned and base proxy models $\boldsymbol{\ell}_{\text{expert}} - \boldsymbol{\ell}_{\text{anti-expert}}$. In our experiments, we proxy-tune a strong base model, LLAMA-3.1-8B, using OLMO2-1B-INSTRUCT and OLMO2-1B as the expert and anti-expert, respectively, which together represent a strong post-training recipe OLMo et al. (2024); Lambert et al. (2025).

Shown in Table 8, we find that the proxy-tuned LLAMA 3.1 (Team, 2024b) model consistently outperforms the base model alone as well as the small tuned expert. This highlights a practical application of ByteSampler to "apply" post-training to base models without actually training them, thus disentangling the quality of the base model from that of the post-training recipe.

| Task | Metric | LLAMA3.1 | OLMO2 INST. | LLAMA3.1 (Proxy Tuned) |
|------|--------|----------|-------------|------------------------|
| AlpacaEval 2 | LC winrate | $0.88 \pm 0.18$ | $\mathbf{33.5} \pm 0.9$ | $\mathbf{33.5} \pm 0.9$ |
| GSM8K | 5 ICE, CoT, EM | $55.3 \pm 2.6$ | $51.9 \pm 2.7$ | $\mathbf{76.6} \pm 2.2$ |
| MMLU | 0 ICE, CoT, MC | $27.8 \pm 0.7$ | $35.2 \pm 0.8$ | $\mathbf{59.5} \pm 0.8$ |

Table 8: **Proxy tuning results.** We report performance on downstream evaluations (with 95% confidence intervals) when proxy-tuning LLAMA3.1-8B using OLMO2-1B-INSTRUCT as the expert and OLMO2-1B as the anti-expert. We see that the proxy tuned model gains the instruction-following capability (AlpacaEval 2) and chain-of-thought capabilities (GSM8K, MMLU) of OLMO2-1B-INSTRUCT while also benefiting from its larger size, allowing it to surpass the expert's individual performance. For details regarding the evaluation, see Appendix B.3.

## 5 CONCLUSION

In this work, we introduced ByteSampler, an algorithm that eliminates the Prompt Boundary Problem by converting any BPE tokenizer-based language model into a byte-level model while preserving its generative distribution at the text-level. Interesting extensions of this method include automatic support for arbitrary pretokenizers (discussed in Appendix C.3), generalization to other tokenization schemes (such as Unigram (Kudo & Richardson, 2018), Wordpiece (Schuster & Nakajima, 2012), and other variants of BPE (Provilkov et al., 2020; Chizhov et al., 2024)), and speculative-decoding at the byte-level.

Beyond correcting sampling artifacts at the prompt-boundary—which is useful in its own right in many situations—the ability to unify vocabularies at inference time enables many forms of model composition, including ensembles of (and post-training transfer between) models with different tokenizers. Other applications of this technology include *(i)* byte-level knowledge distillation to transfer skills more effectively between models with different tokenizers, *(ii)* rapid post-training research leveraging the fact that a post-training recipe (represented by a pair of proxy-tuning experts) can be applied to any number of models without additional training, *(iii)* routing dynamically between models (Zheng et al., 2025) during generation without requiring matching tokenizers, and potentially *(iv)* more convenient LM-powered compression of byte streams.

In general, whenever (mismatching) tokenizers represent an obstacle or inconvenience, our method has the potential to completely bypass it at the cost of (minimally) increased inference compute. We hope that this will prove useful to LM researchers and users alike.

## 6 REPRODUCIBILITY STATEMENT

We include the implementation of our method in the supplementary material (which includes instrumentation required to calculate performance metrics) and all of our results use models and datasets which are publicly available. We also include the

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

## A   RELATED WORK

**Byte-level language models**   Although our method is able to convert a model using a traditional BPE tokenizer into a byte-level model, allowing it to be used in situations where byte-level models are required, it may not enjoy the benefits of being trained natively at the byte level. Training byte-level models are an active area of research (Clark et al., 2022; Xue et al., 2022; Wang et al., 2024). However, byte-level language models may still implicitly aggregate multiple bytes into a single "patch" to help reduce the required sequence length. These patches can be segmented either statically (Tay et al., 2022; Yu et al., 2023) or dynamically (Nawrot et al., 2023; Pagnoni et al., 2024; Ahia et al., 2024) which *may* lead to issues analogous the Prompt Boundary Problem at the patch level, depending on the architecture.

**Tokenizer transfer**   Methods to adapt a model to use tokenizers other than the one they are trained with have been proposed. These methods may rely on interventions during training (Chen et al., 2023), continued training for a subset of the model with the new tokenizer (Marchisio et al., 2023), using self-distillation (Minixhofer et al., 2025), careful initialization of a new embedding matrix, followed by fine-tuning (Minixhofer et al., 2022; Gee et al., 2022; Tran, 2020; Liu et al., 2024e; Dobler & De Melo, 2023), or zero shot transfer using a hypernetwork (Minixhofer et al., 2024). While these methods can, in principle, be used to convert any model into a byte-level model, they will inevitably introduce some distortion into the model's sampling distribution.

**Ensembles of language models**   Many methods to address the mismatching vocabularies one counters when ensembling models have been proposed. These include bridging the vocabularies using a mapping based on model features Huang et al. (2024) or edit distance (Mavromatis et al., 2024) as well as sampling from the union Yu et al. (2024) or intersection Xu et al. (2024b) of multiple vocabularies. There are also several methods that sample multiple tokens of continuation from each model and then select the best one using a scoring metric (Liu et al., 2024d; Xu et al., 2025; Lv et al., 2024). For a survey of such methods, including ones that require training or additional data, see Chen et al. (2025). However, unlike our exact method, all of these methods may introduce distortion into the model's outputs.

**Word level probabilities**   The popular decision to include whitespace with the following word in most modern tokenizers presents a challenge when computing the next word probability (Oh & Schuler, 2024; Pimentel & Meister, 2024), which is closely related to the Prompt Boundary Problem.

**Nondeterministic tokenizers**   Our analysis crucially relies on the determinism of BPE, however nondeterministic tokenizers such as Unigram (Kudo, 2018) and BPE dropout (Provilkov et al., 2020) are of interest to the community. Lundberg (2023) remarks that nondeterministic tokenizers may reduce the severity of the prompt boundary problem, but it cannot do so perfectly. It is possible that more advanced techniques may be able to fully correct the PBP for these tokenizers as well.

## B   EXPERIMENTAL DETAILS

In this appendix, we report additional experimental details.

### B.1   CALCULATION OF THE NAIVE METHOD

The naive method is simple to state. We merely report the average probability that the next character sampled after the prompt will be the correct one. However, some complexity arises when considering multibyte characters, which occur occasionally in English text and essentially constantly in Chinese. A multibyte character may correspond to multiple tokens under a byte-level BPE tokenizer, which means that multiple sampling steps may be necessary to form the next character. To handle this properly, we compute the tree of all token sequences which start with the desired character (depicted in Fig. 2b) and score the log-probability of all of its leaves to determine the exact probability that the desired next character will be generated. Note that we do not perform the pairwise pruning in this step, as we describe in Fig. 2c and Section 3.1. It is not strictly necessary, since a single character can be at most four bytes under UTF-8, so the size of the tree will always be small, and omitting the pruning step presents the baseline in the best light.

### B.2   DETAILS FOR ENSEMBLE EVALUATIONS

For the ensemble evaluations we use few-shot prompting with five in-context examples for each query. We choose the few-shot examples randomly to avoid any bias and ensure that the question being tested is not among the examples. We sample the continuation greedily and test whether the resulting text contains the correct answer.

1. **Arithmetic** contains simple arithmetic problems (Brown et al., 2020).[9] We use the `2da`, `2dm`, and `2ds` splits for addition, multiplication, and division of (up to) 2-digit numbers.

2. **DROP** contains questions about passages, potentially requiring reasoning over multiple pieces of information in the passage (Dua et al., 2019).

3. **Jeopardy** contains open-ended questions from the "Jeopardy!" quiz show (Tunguz, 2019).

4. **LAMBADA** contains narratives without the last word, which is inferrable given the context (Paperno et al., 2016). This task requires models to attend to the full narrative instead of only the local context.

5. **SQuAD** contains passages paired with questions about the passage (Rajpurkar et al., 2016). The answer is always a span from the passage.

6. **TriviaQA** contains open-ended questions about world knowledge (Joshi et al., 2017).

7. **BIG-bench WikidataQA** require models to complete factual statements with the correct continuation (BIG-bench, 2023).

To save compute, we randomly subsample large datasets down to 5,000 examples.

### B.3   DETAILS FOR PROXY-TUNING EVALUATIONS

Following Liu et al. (2024c), we use the proper instruct template for OLMO2-INSTRUCT and use a basic Question/Answer format for the base models. Unlike in the previous section, we use a more varied evaluation setup.

---

[9] https://huggingface.co/datasets/EleutherAI/arithmetic

1. For **AlpacaEval 2**, we prompt using the instruction as the question and take the response as the answer. This is done with no chain of thought prompting or in-context examples. We use the default AlpacaEval 2 judge and report the length-controlled win-rate in our results.

2. For **GSM8k**, we use five in-context examples, which naturally cause the model to produce chains of thought. We extract the final number produced by the model and test if it exactly matches the answer (removing any commas).

3. For **MMLU**, we use no in-context examples and use the chain-of-thought prompt from Lambert et al. (2025) to elicit chains of thought resulting in a multiple-choice answer. Unlike with the other datasets, we do not truncate MMLU to 5,000 examples since its examples are distributed across various domains. We report the multiple-choice accuracy in our results.

These evaluations were intended to benefit from instruction-following capabilities and general knowledge model performance.

### B.4 COMPUTE RESOURCES

Our experiments were conducted with a variety of computing resources, including Nvidia A40, L40S, and A100 GPUs. Our method only requires one GPU at a time and features minimal memory overhead compared to regular sampling. We estimate that the total compute required to reproduce all of our results is less than 200 L40S hours.

## C IMPLEMENTATION DETAILS AND OPTIMIZATIONS

In this appendix, we report implementation details that improve performance and ensure correctness.

### C.1 INFERENCE OPTIMIZATIONS

To ensure that our method is practical we employ a number of optimizations. In order to quickly compute the Valid Cover Tree, we maintain a cache of token masks which are valid following a given token and a separate cache for masks specifying tokens that begin with certain common byte prefixes. Then given a node of the tree, we can quickly expand it, as described in Algorithm 1 by fetching the relevant masks from both caches and intersecting them on the GPU to find the valid children to add.

When evaluating the probabilities of the leaves of the Valid Cover Tree, we use 4D attention masks (S., 2024) to perform inference for the entire tree in a single query. Additionally, while sampling we use KV-caching to avoid redundant computation. Combining these two techniques can lead to excessive memory usage because tokens corresponding to branches that are ultimately not selected by sampling take up space in the KV cache. To address this, we implement a copying garbage collector for the KV cache which discards such tokens from the cache. Since the GC can be run one layer at a time, its total memory overhead is negligible. When using the GC, the KV cache will store exactly one set of keys and values for each token in the *current* Valid Cover Tree, reducing the memory overhead compared to naive sampling to a constant.

We also implement batching, allowing one to sample multiple sequences of bytes in parallel, which permits better utilization of GPU resources.

### C.2 BYTE-LEVEL VS CHARACTER-LEVEL BPE

Throughout this work, we assume that BPE is carried out at the byte level. However, the alternative, performing BPE at the character level, is also a popular choice. Our method can be extended to character-level BPE merges in a natural manner. In particular, one can perform our method at the character level instead. All the analysis we provide, including the guarantees for the Valid Cover Tree in Section 3.1 continue to hold regardless of the choice of base vocabulary. The only additional logic that needs to be implemented revolves around the handling of byte fallback, which is a feature that allows the tokenizer to represent characters that were not included in the base vocabulary explicitly using their Unicode encoding. To handle this properly, we will need to "reset" the tree whenever we encounter a character encoded using byte fallback, since BPE merges do not interact with byte fallback (essentially the byte encoded character acts as a pretokenization boundary). In order to condition on an arbitrary *byte* sequence, we must consider the possibility that a partial character will be completed to form one not in the base vocabulary, necessitating the addition of a "byte fallback"

branch to the Valid Cover Tree. In all other regards, the approach is the same as the one we outline in Section 3.

## C.3 HANDLING PRETOKENIZATION

So far, we have focused on correctly handling byte pair encoding, ignoring the pretokenization conventionally applied beforehand. To illustrate why this is step is important, recall that pretokenization is often used to ensure that tokens cannot span multiple words and that whitespace separating words is merged with the following word and not the preceding one. In order to correctly handle all aspects of modern tokenizers, we must also perform pretokenization in an online fashion, which is challenging in its own right.

Pretokenization is typically implemented using a regular expression: beginning at the start of the text, the longest prefix matching the regular expression is found greedily. This prefix is then extracted into a pretoken and the process is repeated on the suffix. This continues until the entire string has been processed. In order to properly handle pretokenization, we must also perform this splitting online. Due to the expressivity of regular expressions, this requires maintaining a tree of possible splits, which are resolved once enough text is observed, to conclude whether the regex will match or not.

### C.3.1 GENERAL SOLUTION

In principle, the implementation of this idea is straightforward. We can convert any splitting regular expression into a finite automaton, which allows us to detect matches incrementally. By performing connectivity analysis on the automata's state graph, we can infer *(i)* whether there exists a suffix that could produce a regex match (which would mean that the pretokenization might not end up splitting at this point) and also *(ii)* whether there exists a suffix which would cause the regex to stop matching at this point (which would mean that the pretokenization might end up splitting at this point). This analysis can be precomputed for each state in the automaton, allowing these checks to be performed in constant time for each new byte.

If the verdict is ambiguous (both splitting and not splitting are possible), then we add an additional subtree to the Valid cover Tree which assumes that the split has indeed happened. The portion to the left of the split can only be tokenized one way (since its endpoint is fixed), while the portion to the right of the split will be isomorphic to a new Valid Cover Tree for just the portion of the prefix following the hypothetical split. As we continue to add new bytes, we maintain both branches of the tree, just as we would normally. Once enough bytes are added, we can determine conclusively which option was taken, allowing us to discard the subtree corresponding to the opposite possibility.

Of course, it is possible that a new position may occur where the splitting cannot be determined conclusively before the first one is resolved. This will necessitate further splitting of the tree (potentially in both subtrees). In general, this may lead to trees of exponential size, but for typical pretokenizers in use today, we can still guarantee that the tree will have finite size.

### C.3.2 PRACTICAL SOLUTION

Unfortunately, the general solution we outlined in the previous section is difficult to implement in practice. First, most regular expression engines in use today support matching features that are not strictly regular, which makes the conversion of its regexes into automata impossible in the general case. While these features are not used by any pretokenizer we are aware of, the possibility thereof has made it difficult to find routines that are able to perform this conversion for existing regex engines.

To provide a correct implementation while avoiding the complexity of writing our own regex engine, we provide bespoke handlers which are able to handle the pretokenization rules in common use. In general most pretokenization regular expressions have the desirable property that any prefix of a match is also a match. We call this property *closed under prefix*. This makes the detection of possible splitting points very easy, since once the regex stops matching new characters, we know there is no suffix that can extend it. There are only a handful of rules which do not have this property:

- Most tokenizers have a lookahead rule which stops matching whitespace one before the last whitespace. Thus given three spaces in a row, followed by a letter, the first two spaces would be one pretoken and the last space and letter would form a second pretoken.

- Many tokenizers have a "contraction merging" rule which forces contraction suffixes such as ⟨'ve⟩ to be individual pretokens. This is tricky because ⟨'ve⟩ is considered a match but ⟨'v⟩ is not.

We provide handlers for expressions that are closed under prefix, as well as the two special cases we listed above. This is enough to correctly support all pretokenizers we are aware of. (See Appendix C.6 for a list of models tested.)

## C.4 Handling Special Tokens

Special tokens are tokens that are assigned special meaning and are not used simply to represent text. These tokens can have a variety of uses, including marking the beginning or end of documents or separating turns in a dialog. It is easy to handle special tokens in the prompt: when we see a special token, we terminate the tree at that point (discarding potential continuations) and output it, output the ID of the special token, and then start a new tree.

To handle special tokens in the output, we consider the special token distribution on the branch of the tree that ends exactly at the end of the prompt and add those tokens as generation options with the corresponding probabilities alongside the 256 possible next bytes. When composing multiple models which have different sets of special tokens, we require a mapping to specify which tokens have the same meaning. This mapping is automatically detected for BOS and EOS tokens, but must be manually specified for others.

## C.5 Converting Merge Lists to Normal Form

Throughout this work, we have assumed that the tokenizer is constructed using the BPE training algorithm, which proceeds by iteratively merging the most frequent pair in the partially tokenized training corpus. This assumption leads to merge lists that have three desirable properties: *(i)* every token has a unique merge that forms it, *(ii)* every token can be achieved as the tokenization of some string, and *(iii)* the merges always appear in the order they are merged. We assume that these properties are true in the analysis we present in Section 3.

However in practice, some models use "BPE" tokenizers with merge lists that are not directly produced by the BPE training algorithm. One example of this is the tokenizer of LLAMA 3 Team (2024a), which appears to be constructed by extending OpenAI's cl100k tokenizer (OpenAI, 2023) with additional merges intended to add multilingual capabilities. Because of way this extension is done, the LLAMA 3 tokenizer does not have any of the three properties we outlined above. Despite this, inference with the tokenizer is still possible because some tokenization libraries such as HuggingFace Tokenizers[10] employ a heap-based algorithm which simply applies the earliest merge available until no more merges can be applied, which permits the merges to be applied out of order.

Fortunately, it happens to be the case that every merge list can be converted into a functionally equivalent one in "normal form" which has identical behavior while also satisfying the three properties above. This is done using a two step process: *(1)* for each token, we run the heap-based algorithm on it as a string and track which merges are used during the tokenization process. If the resulting token sequence is *not* just the single corresponding token id, then we mark the token as "unreachable" and drop it (this ensures property *(ii)*). Otherwise, we check which merge was applied last and drop any other merges which form the same token since they are also unreachable (this ensures property *(i)*). Then *(2)*, for every merge we check the position of the merges forming its two inputs and move it to immediately after the later of the two if it appears after the original merge (this ensures property *(iii)*). This procedure allows our method to be used with any tokenizer that can be specified using a merge list, even if it was not trained using BPE.

## C.6 Model Support

In Table 1 we sort tokenizers into two primary categories: SentencePiece BPE and HuggingFace ByteLevel BPE. These categories cover the vast majority of modern models, including (but not limited to):

1. SentencePiece BPE: Llama 1/2 (Touvron et al., 2023a;b), Mistral (Jiang et al., 2023), Yi (Young et al., 2024)

---

[10] https://github.com/huggingface/tokenizers

2. HuggingFace ByteLevel BPE: GPT-OSS (Agarwal et al., 2025), Llama 3/4 (Meta, 2024; Team, 2025a), DeepSeek V1/V2/V3/R1 (Bi et al., 2024; Liu et al., 2024a;b; Guo et al., 2025), Qwen 1/2/3 (Bai et al., 2023; Team, 2024e; Yang et al., 2025), GPT-NeoX/Pythia (Black et al., 2022; Biderman et al., 2023), Mistral NeMo (Team, 2024d), OLMo 1/2 (Groeneveld et al., 2024; OLMo et al., 2024), SmolLM 1/2/3 (Allal et al., 2024; 2025; Bakouch et al., 2025), Phi 1/2/3/4 (Gunasekar et al., 2023; Li et al., 2023b; Javaheripi et al., 2023; Abdin et al., 2024), GLM 4 (Zeng et al., 2025), Nemotron H (Blakeman et al., 2025)

3. Neither: Gemma 1/2/3 Team et al. (2024a;b; 2025a), Kimi K2 (Team et al., 2025b), Nemotron 3/4 (Zhang et al., 2024; Adler et al., 2024)

Why does support vary between methods? Beyond implementation details, the methods of Turaga (2025) and Phan et al. (2025) make an assumption about tokenizer behavior (e.g. Proposition 1 in Phan et al. (2025)) that any prefix of a valid token sequence is also a valid token sequence. However HuggingFace tokenizers almost universally use pretokenizers that do not satisfy this assumption.

For example, consider the OLMo 2 tokenizer, which has tokens 220 for "␣" (one space), 256 for "␣␣" (two spaces), and 15 for the digit "0". In this tokenizer we have $\text{encode}(\text{decode}([220, 220])) = [256]$ so $[220, 220]$ is thus invalid. However, we also have $\text{encode}(\text{decode}([220, 220, 15])) = [220, 220, 15]$ so $[220, 220, 15]$ is valid! ByteSampler correctly handles these pretokenizers due to its design (see Appendix C.3).

ByteSampler has been tested all of the listed HuggingFace ByteLevel BPE tokenizers (although not all of the models due to their size) on at least 1 GB of randomly sampled fragments from The Pile (Gao et al., 2020) to ensure the computed Valid Cover Trees are correct.

## D    Technical Details and Proofs

### D.1    Proof of Compactness of the Valid Cover Tree

In this section we prove the following proposition.

**Proposition D.1** (Bounded branching). *Let $S$ be a string and let $T$ be its Valid Covering Tree, as defined in Section 3. Let the trunk of $T$ be the path from the root to the closest node with multiple children. Then the total number of edges in $T$ not contained in the trunk is bounded by a constant $C$ which depends on the tokenizer.*

*Proof.* To bound the size of the tree, we use the observation of Berglund & van der Merwe (2023) that each output token can be fully determined using only a constant amount of lookahead $L$ (in bytes), where $L$ depends only on the tokenizer. This implies that the portion of $T$ excluding the trunk will have depth bounded by $L$. The branching factor of the tree is also bounded by the vocabulary size $|V|$ of the tokenizer. Thus, the number of edges of $T$ is bounded by a constant depending only $L$ and $|V|$. □

### D.2    Proof of Proposition 3.1

First we will prove the following lemma.

**Lemma D.2** (Boundary crossing merges). *Let $S_1$ and $S_2$ be strings and let $T_1 = \text{encode}(S_1)$ and $T_2 = \text{encode}(S_2)$ be their respective token sequences. Then $T_1 + T_2$ is valid if and only if there is no merge applied that crosses the boundary between $S_1$ and $S_2$ while tokenizing $S_1 + S_2$.*

*Proof.* Let $M_1 = m_1^{(1)}, \ldots, m_1^{(n_1)}$ and $M_2 = m_2^{(1)}, \ldots, m_2^{(n_2)}$ be the sequence of merges applied by BPE during $\text{encode}(S_1)$ and $\text{encode}(S_2)$ respectively. Every merge is represented by a tuple $(t, i)$ where $t$ is the index of the merge in the merge list and $i$ is the position in the token sequence where the merge is applied and let merges be ordered lexicographically. Let $M_{12} = m_{12}^{(1)}, \ldots, m_{12}^{(n_1+n_2)}$ be the sorted union of merges $M_1$ and merges from $M_2$ shifted to align with the $S_2$ portion of $S_1 + S_2$ (so each $(t, i) \in M_2$ becomes $(t, i + |S_1|)$ in $M_{12}$).

Let the merge list $M = m^{(1)}, \ldots, m^{(n)}$ be called *invalid* if there exists a merge $m'$ which satisfies either

1. $m^{(i)} < m' < m^{(i+1)}$ and $m'$ matches the partially merged token sequence after $m^{(i)}$,

2. $m' < m^{(1)}$ and $m'$ matches the initial token sequence, or

3. $m^{(n)} < m'$ and $m'$ matches the final token sequence.

If no such merge exists, the merge list is *valid*. If a merge list is valid, then it represents exactly the sequence of merges chosen by BPE, since the BPE algorithm takes the lexicographically smallest available merge at every step. Therefore, by definition $M_1$ and $M_2$ are both valid merge lists.

Now, if $M_{12}$ is invalid, it must be due to a merge $m'_{12}$ that was not available while tokenizing $S_1$ or $S_2$, otherwise that merge would make $M_1$ or $M_2$ invalid. The only such merges are those that cross the boundary between $S_1$ and $S_2$ in $S_1 + S_2$. The only such merges are those that cross the boundary between $S_1$ and $S_2$ in $S_1 + S_2$.

If $M_{12}$ is valid then we will have $\mathrm{encode}(S_1 + S_2) = T_1 + T_2$ and so $T_1 + T_2$ is valid.

On the other hand, if $M_{12}$ is invalid then BPE will apply the invalidating merge $m'_{12}$ which must cross the boundary. Then, since $T_1 + T_2$ has no token that crosses the boundary it cannot be valid. $\square$

Now, recall the statement of Proposition 3.1,

**Proposition 3.1.** *Let* (encode, decode) *denote a BPE encoder and decoder pair corresponding to some merge list $M$ and vocabulary $V$. We call a token sequence $T = [t_1, t_2, \ldots, t_n] \in V^n$ valid if* $\mathrm{encode}(\mathrm{decode}(T)) = T$. *Then $T$ is valid if and only if $[t_i, t_{i+1}]$ is valid for all $i \in \{0, \ldots, n-1\}$.*

*Proof.* If $[t_i, t_{i+1}]$ is valid for all $i \in \{0, \ldots, n-1\}$, then we can show $[t_1, \ldots, t_n]$ is valid by induction. Clearly $[t_1]$ is valid by definition. Now if $[t_1, \ldots, t_i]$ for $i < n$ is valid, then from Lemma D.2 and the validity of $[t_i, t_{i+1}]$, we know there can be no merge the crosses the boundary between $[t_1, \ldots, t_i]$ and $[t_{i+1}]$. This is because a merge can span at most two tokens, so adding more tokens to the left cannot make a new boundary crossing merge possible. Then since $[t_{i+1}]$ is valid by definition, $[t_1, \ldots, t_{i+1}]$ is valid by Lemma D.2 again.

On the other hand, if there exists an $i$ such that $[t_i, t_{i+1}]$ is not valid, then by Lemma D.2 there must exist a merge that crosses the boundary between $\mathrm{decode}\, t_i$ and $\mathrm{decode}\, t_{i+1}$ when encoding $\mathrm{decode}\, t_i + \mathrm{decode}\, t_{i+1}$. This merge can also be applied when encoding $S_1 + S_2$, so by Lemma D.2 again, $T_1 + T_2$ cannot be valid. $\square$

## D.3  DIFFERENCES IN EXACT METHODS

In this work, we consider a method exact if it samples according to the distribution in Eq. (2) modulo the probability mass placed on invalid sequences, which we defined in Section 3.1. Here we describe exactly how these methods differ in their handling of invalid sequences. The method of Turaga (2025) and our method condition on a valid covering of the prompt. This corresponds to the distribution

$$\mathsf{P}\left(t_1, \ldots, t_n \;\middle|\; \begin{array}{l} \mathrm{prompt} \sqsubseteq \mathrm{decode}(t_1, \ldots, t_n), [t_1, \ldots, t_k] \text{ is valid} \\ \text{where } k = \min\{i \mid \mathrm{prompt} \sqsubseteq \mathrm{decode}(t_1, \ldots, t_i)\} \end{array}\right). \tag{3}$$

While difficult to notate, this simply means that the portion of the sequence overlapping the prompt is required to be valid. This is roughly similar to common practice described in Eq. (1) of directly tokenizing the prompt and sampling a continuation while avoiding the PBP. Meanwhile, Phan et al. (2024) consider a relaxation of the above, which does not require the last pair to be valid. This corresponds to

$$\mathsf{P}\left(t_1, \ldots, t_n \;\middle|\; \begin{array}{l} \mathrm{prompt} \sqsubseteq \mathrm{decode}(t_1, \ldots, t_n), [t_1, \ldots, t_{k-1}] \text{ is valid} \\ \text{where } k = \min\{i \mid \mathrm{prompt} \sqsubseteq \mathrm{decode}(t_1, \ldots, t_i)\} \end{array}\right). \tag{4}$$

The less strict conditioning explains why this method has greater overhead, as seen in Section 4.1.

It may seem desirable to sample from the distribution

$$\mathsf{P}(t_1, \ldots, t_n \mid \mathrm{prompt} \sqsubseteq \mathrm{decode}(t_1, \ldots, t_n), [t_1, \ldots, t_k] \text{ is valid}), \tag{5}$$

where the entire sequence is required to be valid. However, it is not clear how to efficiently sample from this distribution. Vieira et al. (2025) highlights this difficulty and propose several alternative

approaches, including approximations of Eq. (5) and architectural modifications that make it easier to sample from Eq. (5).

When applying ByteSampler iteratively, the validity of the sequence is enforced continuously. Since this is done locally, the resulting distribution corresponds to the "locally canonicalized approximation" of Eq. (5) described in Vieira et al. (2025) additionally conditioned on a prompt.

### D.4 SIGNIFICANCE OF INVALID SEGMENTATIONS

For the most part, we have ignored the contribution of invalid sequences to the language model's distribution. This is done out of necessity, since the number of invalid sequences scales exponentially with the prompt length (Vieira et al., 2024). However it is worth considering whether these segmentations could contribute meaningfully to the model's capabilities.

This is closely related to the concept of *marginalization* Cao & Rimell (2021): the idea that calculating the probability of generating a string with a language model requires summing over all segmentations of the string, (including invalid ones). Of note, Chirkova et al. (2023) found that $\mathsf{P}([t_1,\ldots,t_n]$ is not valid) makes up a negligible fraction of the language model's distribution, however later works (Geh et al., 2024; Vieira et al., 2025) came to the opposite conclusion.

### D.5 PROOFS OF EXACTNESS

First we show that the prefix probability calculation is exact.

**Proposition D.3** (ByteSampler prefix probability exactness). *Given a byte-string $P$ with VCT $T$. Let $l_1,\ldots,l_\ell$ be the leaves of $T$ and let $p_i^{(1)},\ldots,p_i^{(m_i)}$ denote the path from the root of $T$ to $l_i$, then*

$$\mathsf{P}(P \sqsubseteq \mathrm{decode}(t_1,\ldots,t_n)) \cong \sum_{i=1}^{\ell} \mathsf{P}(p_i^{(1)},\ldots,p_i^{(m_i)})$$

*where $\cong$ denotes equivalence up to the probability mass placed on invalid token sequences.*

*Proof.* We expand the desired probability in terms of the prefix cover of Vieira et al. (2024) and then remove the invalid sequences

$$\mathsf{P}(P \sqsubseteq \mathrm{decode}(t_1,\ldots,t_n))$$

$$= \sum_{n=1}^{\infty} \sum_{t_1,\ldots,t_n} \mathsf{P}(t_1,\ldots,t_n) \mathbb{1}\left[ \begin{array}{l} P \sqsubseteq \mathrm{decode}(t_1,\ldots,t_n) \\ P \not\sqsubseteq \mathrm{decode}(t_1,\ldots,t_{n-1}) \end{array} \right]$$

$$\cong \sum_{n-1}^{\infty} \sum_{t_1,\ldots,t_n} \mathsf{P}(t_1,\ldots,t_n) \mathbb{1}\left[ \begin{array}{l} P \sqsubseteq \mathrm{decode}(t_1,\ldots,t_n) \\ P \not\sqsubseteq \mathrm{decode}(t_1,\ldots,t_{n-1}) \\ (t_1,\ldots,t_n) \text{ is valid} \end{array} \right]$$

$$= \sum_{i=1}^{\ell} \mathsf{P}(p_i^{(1)},\ldots,p_i^{(m_i)}).$$

The last line follows from the definition of the Valid Covering Tree in Section 3. $\square$

The correctness of the completion sampling and byte level sampling follow because every valid sequence that begins with the prefix must begin with a sequence from the Valid Covering Tree, and Proposition D.3 shows that we have properly accounted for every (valid) sequence that overlaps the prompt.

## E ADVANCED DECODING METHODS

In Section 3, we focused on showing that our method is "exact." To be precise, this means that sampling bytewise using our method and sampling normally give exactly the same distributions of output text (modulo invalid token sequences, as we discussed in Appendix D). However an important distinction arises when applying popular decoding techniques such as greedy decoding, top-$k$, top-$p$ (Holtzman et al., 2020), or even temperatures other than 1. Applying these at the byte-level does not produce the same result as applying them at the token level. This is because these transformations

have different effects when applied with different granularities (clearly, greedily selecting the most likely next byte is not the same as greedily selecting the most likely next token).

When sampling with ByteSampler, we can apply greedy decoding, top-$k$, top-$p$, and temperature, at the byte-level using the normal method. However we can also apply it at the token level by transforming the logprobs of the Valid Cover Tree *prior* to the aggregation into byte probabilities. This preserves exactness in these settings and is able to support any kind of transform that can be applied to the log-probabilities produced by the model.

To explore the difference between these settings, we repeat the code completion experiments from Appendix F.2 using different sampling parameters.

| Sampling transform | Transform level | pass@1 | Avg. completion length |
|---|---|---|---|
| Greedy | Byte | 0.824 | $163 \pm 191$ |
| Greedy | Token | 0.820 | $165 \pm 194$ |
| Top-$p = 0.95$ | Byte | 0.800 | $170 \pm 193$ |
| Top-$p = 0.95$ | Token | 0.787 | $163 \pm 186$ |
| Temperature = 0.7 | Byte | 0.810 | $164 \pm 189$ |
| Temperature = 0.7 | Token | 0.790 | $162 \pm 181$ |

Table 9: **HumanEval Random Span (prefix only)** results using ByteSampler with various sampling parameters

Interestingly, we find that the byte-level sampling transforms tend to slightly outperform the corresponding token-level sampling transforms. We think exploring the cause of this difference in performance is an interesting direction for future work.

## F  EXTRA EXPERIMENTAL RESULTS

In this appendix, we report additional experimental results that did not fit in the main text.

### F.1  DETAILED PERFORMANCE COMPARISON WITH PHAN ET AL. (2025)

Comparing the performance of ByteSampler with that of Phan et al. (2025) is difficult because there is no model that is supported by both methods. In this section we make an approximate comparison using OLMo 2 7B with our method and Llama 2 7B with the method of Phan et al. (2025). These models have very similar size (7.30B and 6.74B respectively) and both use a standard dense transformer architecture without employing multi-query attention or grouped-query attention. Our benchmark setting is to sample 100 byte completions to questions from MMLU.

| Method | Model | Inference tokens | Throughput |
|---|---|---|---|
| Phan et al. (2025) | Llama 2 7B | $637 \pm 518$ | $13.8 \pm 0.4$ bytes/sec |
| ByteSampler (ours) | OLMo 2 7B | $\mathbf{190} \pm 177$ | $\mathbf{37.1} \pm 0.8$ bytes/sec |

Table 10: **Performance metrics for ByteSampler and the method of Phan et al. (2025)**. We report the total number of model inference tokens consumed by the method as well as the average number of bytes generated per second for each method. ByteSampler is significantly faster and requires fewer inference tokens. OLMo 2 7B is slightly larger than Llama 2 7B, so we expect ByteSampler to be at a slight disadvantage in this comparison.

### F.2  CODE COMPLETION

To demonstrate the importance of the prompt boundary problem to longer generative tasks, we measure performance on HumanEval Random Span (Bavarian et al., 2022), a FIM ("Fill in the Middle") code-completion variant of HumanEval (Chen et al., 2021). The dataset is made of examples with a prompt and a suffix. The original task given a prompt and suffix is to generate a string middle such that the code prompt + middle + middle will pass the tests. To make this task more suitable for our setup, we discard the suffix and ask the model to directly extend the

prompt into a valid solution. The in HumanEval Random Span, the prompt is selected to include the instructions for the code as well as a random prefix of the solution code itself, which makes this setting likely to exhibit prompt boundary issues. We report the results for Qwen3 1B in Table 11.

| Method | pass@1 |
|---|---|
| Naive | 0.565 |
| 1 Token Backtracking (Token Healing) | 0.716 |
| 1 Token Backtracking (Token Healing) | 0.741 |
| 1 Token Backtracking (Token Healing) | 0.738 |
| ByteSampler (ours) | **0.824** |

Table 11: **HumanEval Random Span (prefix only)** results for naive sampling, backtracking, and ByteSampler.

## G    EXAMPLE VALID COVER TREES

Here we show complete Valid Cover Trees for several example prefixes. Unlike the tree in Fig. 2c, we show the actual tree as calculated by our algorithm. However to allow them to fit on a page, we choose to display *only the internal nodes* of the tree (not the leaves). To denote where the hidden leaves would be, we display nodes that have leaves in **bold font**.

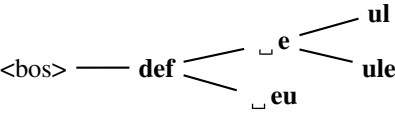

Figure 4: Example Valid Cover Tree for prefix "this is a tes" with the OLMO 2 tokenizer.

Figure 5: Example Valid Cover Tree for prefix "def eule" with the OLMO 2 tokenizer.

We hide the leaves because it is typical for nodes that do have leaves to have dozens or even hundreds of them. To see how this can occur, imagine a prompt that ends on a space, and an internal node that ends right before that space. The node's children will be all valid tokens that begin with a space. Most tokenizers have tens of thousands of tokens which begin with a space and nearly all of them will be valid continuations.

While this may sound problematic, we only need to query the next token distribution for the parent once in order to score all of its children, so this can be done efficiently in combination with the masking cache we describe in Appendix C.1.

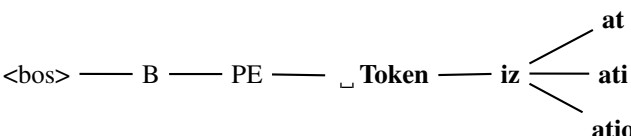

Figure 6: Example Valid Cover Tree for prefix "BPE Tokenizatio" with the OLMO 2 tokenizer.

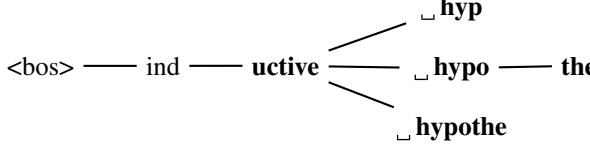

Figure 7: Example Valid Cover Tree for prefix "inductive hypothe" with the OLMO 2 tokenizer.

<bos> —— 日本 —— 的 —— 首 —— 都是 —— 东京 —— ， —— 中国的 ⟨ 首 🌿 / 首都 🌿

Figure 8: Example Valid Cover Tree for prefix "日本的首都是东京，中国的首都" with the QWEN3 tokenizer. We use 🌿 to denote nodes with leaves omitted.

