# OpenReview forum: "Sampling from Your Language Model One Byte at a Time"
_ICLR.cc/2026/Conference — Submitted to ICLR 2026_

### Official Review · Reviewer_Qz8v · 2025-10-30

**Soundness:** 3
**Presentation:** 3
**Contribution:** 2
**Rating:** 4
**Confidence:** 4

**Summary:**

This paper presents ByteSampler, a method of treating a distribution over token sequences (an LLM) as the corresponding distribution over character strings, assuming that the probability of non-canonical token sequences in the LLM is 0. It provides three capabilities given a prompt $S$ in the form of a string of characters/bytes, as summarized in Section 3.2: (1) compute the total probability that $S$ is a prefix of a canonical token sequence generated by the LLM; (2) sample a completion of $S$ while avoiding the prompt boundary problem; and (3) compute the probability distribution over next characters/bytes (rather than tokens) following $S$, providing the ability to convert a token-level LLM to a character/byte-level one. The method works by constructing a tree of canonical token sequences where the final token straddles the boundary of $S$. The experiments show that the byte-level interface results in comparable bits per character on naturalistic data to the original token-level LLM while not adding too much overhead in terms of calls to the underlying token-level LLM. Being able to convert a token-level LLM also adds the ability to ensemble LLMs with different tokenizers or proxy-tune models with different tokenizers; and the authors present fairly strong results on these applications.

**Strengths:**

The method provides a sensible solution for the prompt boundary problem and incurs relatively little computational overhead, so it clearly has practical use. The paper is generally clear and easy to follow. The idea of converting token-level LLMs to character/byte-level ones to enable ensembling and proxy tuning of models with different tokenizers is exciting and opens up a lot of new possibilities. With the exception of TriviaQA and WikidataQA in Table 7, the experimental results in Section 4.3 and 4.4 are quite strong. The paper includes an excellent breadth of models and benchmarks.

**Weaknesses:**

This paper has quite a bit of overlap with two recently published papers, which I will refer to as [1] and [2]:

* [1] Vieira et al. (2025). [From Language Models over Tokens to Language Models over Characters](https://openreview.net/forum?id=sQS0roNQZR)
* [2] Vieira et al. (2025). [Language Models over Canonical Byte-Pair Encodings](https://openreview.net/forum?id=eCVrfVDNSY)

[1] also provides algorithms for converting a token-level LLM to a character/byte-level one, and [2] provides algorithms for enforcing the canonicality of BPE token sequences.

1. The characterization that ByteSampler provides an "exact" solution to the prompt boundary problem is misleading. Like this paper, [1] also provided algorithms for converting a token-level LLM to a character/byte-level one. The beam summing algorithm of [1] (approximately) marginalizes over all tokenizations that match a given character sequence, including non-canonical ones. This corresponds closely to this paper's Eq (1) (incidentally, the equation seems to be missing the probability of EOS). Crucially, ByteSampler relies on the assumption that the probability of non-canonical token sequences in the LLM is negligible. However, [2] showed that, in practice, this is often not the case. For example, LLaMa 1B and LLaMa 8B have a log-canonicality rate of about -1.2, or roughly 30% -- so about 70% of sampled token sequences are non-canonical. This means that the distribution that ByteSampler induces is considerably distorted from Eq (1) and from the actual distribution over character strings that LLMs generate in practice. The distribution of ByteSampler corresponds more closely to the "locally canonicalized" method of [2] (Section 3.2.2).
1. The most comparable prior work is the beam summing algorithm of [1], but this paper does not sufficiently compare against it. It compares ByteSampler against [1] only in terms of computational cost, not the quality of the distribution over character strings, and it only compares against the exponential-time version that does not use beam search. [1] includes a beam size hyperparameter $K$ that can be used to trade computational cost for fidelity to the true marginal distribution over character sequences, in which case the time complexity is linear in the length of the character string, not exponential. A good comparison would be to show the Pareto frontier of character-normalized cross-entropy on naturalistic data vs. computational cost. Perhaps ByteSampler outperforms [1] for some range of $K$.
1. The pairwise validation for BPE in Proposition 3.1 was already proven and used by [2], as was the trick for checking the merge trajectory along the boundary between the tokens for conflicts. The paper does not acknowledge this.
1. A minor point: In Table 3 and similar tables, the values in the loss per unit column are not comparable, so it doesn't make sense to bold the lowest score.
1. Very minor: Table 6 is not referenced in the main text.

**Questions:**

1. Is Eq (1) missing EOS?
1. Table 3: Why is the bits per character score for ByteSampler bold?
1. 322: How did you get this conversion rate?
1. Table 3: Can you provide the formula for computing bits per character?
1. 328: Are you including the probability of EOS when computing the probabilities of these documents?

---

> ### Author Response · Authors · 2025-11-20
>
> We thank the reviewer for the insightful comments, and for recognizing the “practical use” of ByteSampler to “open up new possibilities” and the “strong experimental results.” We provide discussion on all of your comments and hope that it addresses your concerns.
>
> ### Exactness and invalid token sequences
>
> We are using the same definition of exactness as Phan et al. [3] (and [4] even earlier), so we are **using the term “exact” by convention**. You are correct that this notion of exactness relies on an assumption of zero (or at least negligible) probability mass on invalid token sequences, which does not hold in practice. However, we believe **our definition makes practical sense**: when just solving the prompt boundary problem, our method ensures that all tokens overlapping the prompt must be valid, but places no constraints on subsequently generated tokens. This **closely matches the standard practice of tokenizing the prompt (in the canonical fashion)** and then sampling its completion in an unconstrained manner.
>
> However, as you note, when sampling at the byte-level, the repeated conditioning enforces the validity constraint for generated tokens (those overlapping previously generated bytes) as well. This gives the locally canonicalized distribution from Vieira et al. (2025) [2]. (Phan et. al. give a slightly different distribution.) While it is clear that language models do occasionally generate invalid token sequences, it is not clear whether it is helpful to allow them to do so. Our results for the beam summing algorithm (which does consider invalid token sequences) do not show any clear gain in byte-level perplexity compared to our approach. That said, we believe this is an interesting question and plan to run further experiments along these lines.
>
> [3] Phan, Buu, et al. "Exact Byte-Level Probabilities from Tokenized Language Models for FIM-Tasks and Model Ensembles." *The Thirteenth International Conference on Learning Representations.*
>
> [4] Phan, Buu, et al. "Understanding and Mitigating Tokenization Bias in Language Models."
>
> # Comparison to beam summing algorithm
> Following the reviewer suggestion, we ran new experiments for the beam summing algorithm. (We omitted these experiments initially because we believed the algorithm of Phan et. al. (2025b) [3] would be more efficient *and* exact, but to our surprise, we found this to not always be the case — the beam summing algorithm is faster than Phan et. al. at small beam size!) Because the beam summing algorithm relies on automatic prefix caching in VLLM, it is difficult to tell exactly how many tokens of inference are being used, so we report the algorithm speed in seconds (wall time). For a fair comparison, we rewrote our solution to use the same VLLM-based backend.
>
> First, we measure the average byte-level cross-entropy loss of each method on a dataset of 10,000 document prefixes of 100 bytes from the OLMo 2 training set. We use the Llama 3 1B model because the beam summing implementation does not support OLMo 2 (for reasons unknown to us).
>
> | Method | Loss ↓ (nats/byte) | Speed ↑ (bytes/sec) | Error rate ↓ (%) |
> | :- | -: | -: | -: |
> | ByteSampler | **0.986** ± 0.003 | **620** | **0.0** |
> | Beam Summing (K = 1) | 1.022 ± 0.003 | 153 | 25.6 |
> | Beam Summing (K = 3) | **0.986** ± 0.003 | 118 | 1.47 |
> | Beam Summing (K = 10) | **0.986** ± 0.003 | 48 | 1.42 |
> | Beam Summing (K = 128) | **0.986** ± 0.003 | 4.7 | 1.42 |
>
> Shown in the table, we find that **ByteSampler is significantly faster than even a beam size of 1, while achieving similar loss to much larger beam sizes.** It’s worth noting the beam summing algorithm can error when the beam becomes empty due to a lack of valid continuation tokens; this occurs for 25% of prefixes with using beam size = 1! We have recorded the error rate here, but the loss numbers are based on examples where no method errored.

---

> ### Author Response · Authors · 2025-11-20
>
> # Comparison to beam summing algorithm (cont.)
>
> Additionally, we report generation speed under two different regimes: 1. Sampling heavy and 2. Prompt heavy (prefill).
>
> | Method | Prompt bytes | Sampled bytes | Time ↓ (s) |
> | :- | -: | -: | -: |
> | ByteSampler | 0 | 500 | 14.2 |
> | Beam Summing (K = 1) | 0 | 500 | **13.1** |
> | Beam Summing (K = 3) | 0 | 500 | 15.5 |
> | Beam Summing (K = 10) | 0 | 500 | 21.1 |
> | ByteSampler | 1000 | 100 | **2.95** |
> | Beam Summing (K = 1) | 1000 | 100 | 25.4 |
> | Beam Summing (K = 3) | 1000 | 100 | 29.6 |
> | Beam Summing (K = 10) | 1000 | 100 | 35.2 |
>
> We see that **in the prompt-heavy regime (bottom half), ByteSampler achieves 10x better efficiency**, while achieving efficiency comparable to beam summing with K=1 in the sampling-heavy regime (top half). This makes sense because ByteSampler can prefill text at the transformer’s native speed, while Beam Summing needs to carry the beam state forward through the prefilled text byte by byte, which is much slower. We will be sure to discuss this advantage in the next revision (this is also why ByteSampler is much faster at scoring).
>
> We believe these results strengthen the argument for our solution, as **it is very fast while also achieving the high language modeling quality expected from an exact method.**
>
> ### Prior usage of pairwise validation
> We acknowledge that the pairwise test was used earlier in Antwerpen & Neubeck (2024) [5] in footnote 1 of our paper. We will also clarify that the optimization is the same as the one used in Vieira et al. (2025) [2], although we discovered it independently.
>
> [5] Hendrik van Antwerpen and Alexander Neubeck “So many tokens, so little time: Introducing a faster, more flexible byte-pair tokenizer”
>
> ### EOS in Eq (1)
> For convenience, we are writing the expression for the prefix probability of the string, which does not require the EOS. (We will make this more clear in the revision). Our method does handle EOS correctly and can compute both prefix and complete string probabilities (see Appendix C.4 for a description of how EOS is handled), however we chose to use the prefix probability in our notation to avoid having to special-case the EOS token (which has no byte representation and thus cannot be “decoded” in the standard sense). We will clarify this in the revision.
>
> ### Bold in Table 3
> We bold the result that is the best among all alternatives (i.e. the no mitigation baseline). The “Plain BPE” number is not comparable with the other two because it is evaluated on a different set of prefixes (only whole tokens), however we include it as it gives a useful reference for the numerical values. We will make this point more clear in the revision.
>
> ### Conversion rate and formula
> We used the average character per token rate for the dataset we ran the evaluation on. To avoid unwanted correlations, we calculated the statistics over a separate sample from the same dataset of one million tokens. We will clarify this point in the revision.
> To convert nats/char to bits/char, we multiply by 1/ln(2). To convert nats/token to bits/char, we divide by the token to character rate (e.g. 4.518) giving nats/char, and then convert to bits/char.
>
> ### EOS for documents
> As in Eq (1), we measure the prefix probability (no EOS). This is sensible since we are truncating the documents to begin with (we discard documents that are not long enough to be truncated), so the text itself is best interpreted as a prefix. We will clarify this point in the revision.

---

### Official Review · Reviewer_E5BB · 2025-10-31

**Soundness:** 3
**Presentation:** 3
**Contribution:** 3
**Rating:** 6
**Confidence:** 4

**Summary:**

This paper studies the problem of compute next-byte probabilities for autoregressive sampling from a tokenized LLMs. The authors introduces a construction of valid covering tree that improves robustness and speed over prior works. For applications, they show the advantage of this byte-sampler procedure on fixing broken tokens, model ensemble and proxy tuning.

**Strengths:**

The application of byte-level sampling to proxy tuning is interesting and should be further emphasized as a key contribution of the paper.

The proposed method demonstrates clear improvements over prior approaches in both speed and robustness.

Moreover, the comparison between the byte-sampling strategy and the token-healing approach is well-presented and helps clarify the advantages of the proposed technique.

**Weaknesses:**

There are notable overlaps between the proposed method and prior work such as [1], which also introduces a similar byte-level sampling procedure and targets applications like handling broken tokens in code and ensembling LLMs. Nevertheless, [1] also has certain limitations, particularly that its pre-tokenization handling is restricted to older BPE-based tokenizers (e.g., SentencePiece), as detailed in Section C.6 where this paper also provides solution to handle such problem. In my opinion, this distinction should be emphasized more clearly in the main paper rather than being overshadowed by the ensembling experiments, so that readers without a strong tokenization background can better appreciate the robustness advantage. For this reason, it might also strengthen the paper to revise the title to highlight this aspect, for example: “Robustly Sampling from Your Language Model One Byte at a Time.”

The writeup in Section 3.2 is not very well-organized. Perhaps providing an example such as on a simple Markov chain would improve the clarity. Also, equation (2) needs more explanation so that it lines up with prior works [1,2].

[1] Buu Phan, Brandon Amos, Itai Gat, Marton Havasi, Matthew J Muckley, and Karen Ullrich. Exact byte-level probabilities from tokenized language models for fim-tasks and model ensembles. In The Thirteenth International Conference on Learning Representations, 2025.

[2] Tim Vieira, Ben LeBrun, Mario Giulianelli, Juan Luis Gastaldi, Brian DuSell, John Terilla, Timothy J O’Donnell, and Ryan Cotterell. From language models over tokens to language models over characters. arXiv preprint arXiv:2412.03719, 2024.

**Questions:**

Could you elaborate on the differences among the methods discussed in Section D.3?

Regarding proxy tuning, would it be possible to incorporate additional experts or anti-experts to further enhance performance? It would be interesting to include more experimental results regarding this.

In Phan et al. [1], an O(1) procedure for next-byte sampling is also introduced. Setting aside robustness concerns such as pre-tokenization issues, could you clarify how your approach improves sampling speed compared to theirs?

---

> ### Author Response · Authors · 2025-11-20
>
> We thank the reviewer for the thoughtful comments and recognizing that ByteSampler enables computing next-byte probabilities from tokenizer-based LMs with “improve[d] robustness and speed over prior works.”
>
> ### Comparison with Phan et al. [1]
>
> As the reviewer points out, one significant advantage of ByteSampler over Phan et al. is that **we support modern BPE-based tokenizers** by being able to handle their complex pretokenization. Briefly, this is because of the different algorithm structures: ByteSampler builds a tree incrementally left-to-right rather than tokenizing backwards, thus supporting pretokenization that splits in the past based on future bytes.
>
> Besides this, ByteSampler is also **significantly more efficient than Phan et al.** ByteSampler is 3x cheaper in terms of inference tokens (see the per-byte overhead in Table 2 and Table 10 in Appendix F.1). This is due to our more aggressive pruning of invalid token sequences. (We also use 2x less KV cache memory due to our KV garbage collector, although this is not mentioned in the paper.)
>
> With that said, we will emphasize the value of our improved model coverage in the next revision following the reviewer suggestion, as support for a large number of modern models is an important advantage in its own right.
>
> ### Improvements to presentation
> We acknowledge that we are going too quickly through some important information in Section 3.2. We will do our best to improve the presentation. We think the best way to do this may be to move Section 3.1 into the Appendix, which will give us space for examples and figures in Section 3.2 as you suggest.
>
> ### Different approaches to exactness
>
> Essentially, the differences in definitions revolve around where to enforce validity of the token sequence. For example, the standard practice of tokenizing the prompt and then sampling its completion (in an unconstrained manner) enforces validity of the prompt but not the completion. When extending to solutions solving the prompt boundary problem, we need to consider tokens which cross the prompt/completion boundary. In Eq. (3), we require those tokens to be valid (w.r.t the preceding text) as well, while in Eq. (4) we do not. It may seem natural to enforce the validity constraint everywhere, as described in Eq. (5), but this is not computationally tractable.
>
> In terms of downstream performance, we do not expect there to be any substantial differences between Eq. (3) and Eq. (4) (we expect it to fall well within the noise floor for our evals). However, Eq. (3) is faster because the more aggressive pruning means fewer inference tokens are required (see, e.g. Table 2).
>
> ### More expert pairs for proxy tuning
> This is a very interesting idea, which combines proxy tuning with ensembling! We will try to finish it in time for the revision, though our results are not ready yet due to compute constraints. Each of our 1B base models (OLMo 2, Llama 3, and Qwen 3) have a corresponding chat-tuned model, so there are lots of possible combinations for this experiment!
>
> ### Speed improvement over Phan et al. [1]
>
> It is not possible to perform a perfect comparison with Phan et al. [1] as the two approaches support disjoint sets of models. Nonetheless, we attempted to compare them in similar settings in Appendix F.1 and found our method was substantially faster. We believe the improvements in speed are mainly due to the following two advantages:
>
> 1. The 3.3 × reduction in inference tokens, due to the more aggressive pruning using the token validity constraints.
> 2. Reduced preprocessing complexity, due to the incremental construction of the covering tree. (Phan et al. [1] effectively rebuild the tree from scratch for every byte.)

---

> > ### Comment · Reviewer_E5BB · 2025-11-22
> >
> > Thank you for your response. The writing is not super clear to me how your algorithm works as every thing looks very disconnected.
> >
> >  For instance algorithm 1, when do you invoke the model call? How the tree structure improves the robustness?These are not super clear to me. Maybe some diagram would help.

---

> > > ### Author Response · Authors · 2025-11-23
> > >
> > > ### Algorithmic structure
> > > We apologize for the lack of clarity concerning the algorithm. The general structure of our execution is:
> > >
> > > 1. Obtain the valid prefix tree
> > > 2. Score the tree using the model
> > > 3. Perform some aggregation/selection using the resulting scores
> > >
> > > We describe several such routines in Section 3.2, but we did not go into detail due to space constraints (which we plan to address). Here, we have written each of them out in Python-like pseudocode:
> > >  ```
> > > def alg(prefix: bytes, model):
> > >     # Step 1: compute the VCT (Section 3 Introduction, 3.1)
> > >     vct = compute_valid_cover_tree(prefix)
> > >
> > >     # Step 2: score the tree using the model
> > >     leaves_with_logprobs = tree_inference(model, vct) # Single-call using 4D attention masks
> > >
> > >     # Step 3: aggregate/select (examples in Section 3.2)
> > >
> > >     # e.g. to compute the log-prob that model will generate a string starting with prefix:
> > >     prefix_logprob = logsumexp(leaf.logprob for leaf in leaves_with_logprobs)
> > >
> > >     # e.g. to extend the prompt to a point where we can continue it with normal sampling
> > >     healed_prompt = leaf_to_string(sample(
> > >         choices=leaves_with_logprobs,
> > >         logprobs=[leaf.logprob for leaf in leaves_with_logprobs],
> > >     ))
> > >
> > >     # e.g. to get the next byte_distribution
> > >     buckets = {b: [] for b in range(256)}
> > >     for leaf in leaves_with_logprobs:
> > >         # get the byte immediately after the prompt ends
> > >         next_byte = get_implied_next_byte(prefix, leaf)
> > >         buckets[next_byte].append(leaf.logprob)
> > >     next_byte_dist = {b: logsumexp(logprobs) for b, logprobs in buckets.items()}
> > > ```
> > > We intend to expand Section 3.2 to make it more clear (as we outlined previously, our plan is to move Section 3.1 to the appendix to make room and we are open to moving Section 3.3 as well if needed). The key to making this system efficient is to make Steps 1 and 2 incremental, which is done using streaming tree updates (Section 3.3) and KV caching respectively.
> > >
> > > ### Robustness
> > > The main source of lack of robustness of the method of Phan et. al. [1] is due to pretokenization (the splitting of the text into chunks before applying the BPE algorithm). Nearly all tokenizers that descend from GPT 2 (i.e. those of almost all modern models) use pretokenization regexes that split at certain positions conditioned on what text appears later. (We describe an example of this phenomenon in Appendix C.6.) These tokenizers do not satisfy Proposition 1 of Phan et. al. [1], which assumes that the validity of a sequence of tokens does not depend on future tokens.
> > >
> > > Our tree-based approach is able to handle this by explicitly modeling the pretokenization stage. When a pretokenization split is currently indeterminate (depending on future bytes), we create branches in the tree covering both possibilities. Later, once enough bytes have been observed to determine whether the split happened, we can prune the branch that is no longer valid. (See Appendix C.3 for how we do this.)
> > >
> > > This is an important advantage of our method, so we plan to emphasize it more in our revision.

---

### Official Review · Reviewer_Nkha · 2025-11-01

**Soundness:** 2
**Presentation:** 3
**Contribution:** 2
**Rating:** 2
**Confidence:** 4

**Summary:**

This paper presents a method for converting token-level language models to character-level (byte-level) models. The paper claims several contributions. First, an O(1) overhead algorithm for conditioning token-level LMs on character strings, a novel Valid Covering Tree concept for enumerating valid tokenizations, an efficient bigram-based canonicality test for BPE, and empirical validation showing the method achieves low error and improved bits/byte compared to canonical tokenization baselines. ByteSampler does not add high overhead relative to plain BPE decoding while substantially reducing boundary artifacts, and it evades the worst-case exponential cost of unpruned prefix-cover enumeration.

**Strengths:**

The empirical evaluation is comprehensive, testing on four modern LLMs (Llama-3.2-1B, Meta-Llama-3.1-8B, DeepSeek-R1-Distill-Llama-8B, and phi-4) with careful measurement of the speed-accuracy tradeoff. The implementation appears to be well-engineered, with the bundled beam summing approach using trie-based filtering (Appendix D) providing meaningful practical speedups. The pedagogical presentation is generally clear, with helpful visualizations like the Valid Covering Tree diagrams that make the concepts accessible. The paper demonstrates that even with modest computational budgets (small beam sizes), reasonable approximations to the character-level distribution can be achieved, which is useful for practitioners. The experimental design measuring both Jensen-Shannon distance to a reference model and bits/byte compression is informative.

**Weaknesses:**

The core weakness of the paper is lack of novelty:

The algorithm is similar to a previous algorithm from Vieira et al. (2025b). The paper claims throughout to provide exact solutions, but this claim is not clear as the authors redefine what "exact" means to be "modulo invalid token sequences".  The method only ensures canonicality of tokens overlapping the prompt prefix, not the entire sequence, making it an approximation to the ideal distribution. This limitation is in Appendix D.3 rather than discussed in detail. The exactness claim relies on an assumption that noncanonical strings have zero probability, but this assumption doesn't hold according to Geh et al. (2024) and Vieira et al. (2025b).

The complexity comparison in Table 1 is also not clear. It compares ByteSampler's practical O(1) overhead against Vieira et al. (2024)'s theoretical 2^O(n) worst-case bound, while not comparing against the "beam summing algorithm" in the same paper with O(N·K·|∆|) complexity, which is similar..

Additionally, the bigram test is similar to previous work by Antwerpen & Neubeck (2024) and Vieira et al. (2025b).

**Questions:**

Did you measure the probability mass on noncanonical strings in your experiments? If so, did you consider how does considering noncanonical tokenizations affect performance on downstream tasks?

How does the bigram test differ from previous papers on canonicality?

If possible, it would be helpful to test the method to other canonicalization papers to understand the difference in runtime and accuracy, to differentiate the method more from previous work.

---

> ### Author Response · Authors · 2025-11-20
>
> We thank the reviewer for recognizing ByteSampler’s “novel Valid Covering Tree” approach to converting tokenizer-based LMs into byte-level LMs, and that it “[does] not add high overhead… while substantially reducing boundary artifacts.” We hope our response addresses all of the weaknesses and questions raised.
>
> ### Comparison with Vieira et al. (2025b) “Language Models over Canonical Byte Pair Encodings”
>
> While there is some similarity between our work and Vieira et al. (2025b), the two solutions solve different problems. Ours is intended to solve next-byte prediction and the prompt boundary problem, an online problem where future bytes are not known, while Vieira et al. (2025b) is designed to classify which next tokens are valid given the previous token. Our test is the same as the one used in Vieira et al. (2025b) (although we discovered it independently), but it is used in a different way and for a different purpose.
>
>
> That said, our implementation of checking the validity of adjacent tokens is actually more robust than that of Vieira et al. This is because modern tokenizers are more than just BPE merges: they have complex behaviors arising from features such as normalization, pretokenization, “ignore_merges”, added tokens, special tokens, which cannot be described by merge rules alone. To address this, Vieira et al. (2025b) construct a list of overrides, i.e., observed pairs that the BPE test says should not be possible, which are learned from data. This is inherently fragile — if an override is missing, the sampler will break!
>
>
> In contrast, our method **fully characterizes the set of possible valid token sequences** by stacking tree transducers corresponding to each step of the modern tokenization pipeline. This system ensures that every ambiguity in possible tokenization is resolved once enough future bytes have been observed and prevents an exponential blowup due to stacked ambiguity.
>
> ### Probability mass on noncanonical token sequences
> We measure the cross-entropy loss of Vieira et al. (2025b)’s local canonicalization method against the converted cross-entropy loss of normal token level prediction (for the same dataset as in Table 3).
>
> Here we measure the cross-entropy loss of the normal token level model and compare it to the loss when invalid next tokens are masked out (for the same dataset as in Table 3).
>
> | method | loss (nats/byte) |
> | :- | :- |
> | Token level | 1.050322 |
> | Local Canonicalization | 1.050033 |
> | Delta | 0.000288 ± 0.000006 |
>
> This small difference in loss translates to a **high canonicality rate** of 97.15% for 100 byte sequences. We also want to point out that even though the Beam Summing algorithm of Vieira et al. (2024) [1] sums over non-canonical sequences, ByteSampler still achieves the same loss. (See our response to Reviewer Qz8v for details.)
>
> [1] Vieira, Tim, et al. "From Language Models over Tokens to Language Models over Characters." *Forty-second International Conference on Machine Learning.*

---

> ### Author Response · Authors · 2025-11-20
>
> ### Definition of exactness
>
> We use the same definition of exactness as Phan et. al. [2], so our definition is supported by prior work. Of course models do, in-practice, place non-zero probability on invalid tokenizations, so this assumption is not perfect. However, we believe **this definition of exactness makes practical sense**, as it closely matches the standard practice of tokenizing the prompt (in the canonical fashion) and then sampling its completion in an unconstrained manner. Indeed, ByteSampler differs from normal sampling only in that it prevents the model from generating invalid token sequences, and we have no reason to believe that allowing models to commit tokenization errors during sampling is beneficial. While Geh et al. (2024) do find that invalid token sequences contain useful signal, they do so in a targeted manner that is not equivalent to normal sampling and cannot be easily adapted to a general sampling method.
>
> [2] Phan, Buu, et al. "Exact Byte-Level Probabilities from Tokenized Language Models for FIM-Tasks and Model Ensembles." The *Thirteenth International Conference on Learning Representations.*
>
> ### Including Beam Summing algorithm of Vieira et al. (2024) in Table 1
>
> We omitted the beam summing algorithm from Table 1 because it is slower than the algorithm of Phan et. al. while also not being exact (modulo invalid sequences). However, we will add it to Table 1 for completeness. Additionally, we have run several experiments comparing the beam summing algorithm to ByteSampler, which show that our method is much faster than the Beam Summing algorithm while achieving the same language modeling quality (please see our response to Reviewer Qz8v for details).
>
> ### Discussion of bigram test in prior work
> While Antwerpen & Neubeck (2024) [2] also check the pairwise validity of two tokens, they use a brute force approach; we mention this work in footnote 1 of our paper. Our optimization of checking along the boundary is the same as the one used in Vieira et al. (2025b), although we discovered it independently. We will credit Vieira for the independent discovery in our next revision.
>
> [2] Hendrik van Antwerpen and Alexander Neubeck “So many tokens, so little time: Introducing a faster, more flexible byte-pair tokenizer”

---

### Official Review · Reviewer_KhEW · 2025-11-02

**Soundness:** 2
**Presentation:** 2
**Contribution:** 3
**Rating:** 4
**Confidence:** 5

**Summary:**

This paper presents an efficient algorithm to the prompt boundary problem or tokenization bias problem.

Lines of attack are execution speed, memory efficiency and possibly mixed generation, i.e. generating bytes only if there is a tokenization bias expected, otherwise generating tokens.

Note that I do not want to score the paper at this moment I will assign scores after our discussion.

**Strengths:**

Making more efficient versions of exact solution to the PBP is crucial for its adoption in all decoding libraries and ultimately for its broad appeal.

I will ask more clarifying questions I find the writing somewhat unclear.

**Weaknesses:**

I struggle a lot reading this paper. For one the contributions are not clear to me exactly i also think it overclaims here and there and the experiments are a bit poorly chosen to illustrate the contributiin. Let me get more specific ( to be clear nothing you couldnt fix in the camera ready ).

In no specific order

1 in abstract be specific about what efficiecy you improve

2 in contributions i would restrict my claim to the efficency bit bc as u say yourself other work has already provided exact solutiond to the problem + do not claim ensembeling that was already done in prior work or if u do how is yours better

3 u claim exact solutions but never define what that means i assume u refer to phans definition of statistically equivalent u should define what u mean

4 in that context u should discuss clearee what distingushes your work from phan, in line 145

5 in 3 you talk about valid coverings again i must assume u take that inspired by phan that define valid encodings and covers but i dont know u need to define that, or do you mean vieira?

6 proposition 3.1 is not clear the first part is a definition the second one was already shown i phan just cite theirs

7 3.2 and 3.3 i will ask more clarifying questions in discussion

8 experiments: from your method i come out believe you came up with a more efficient version vieira and phan hence experiments should focus on that

9 what is overhead with bpe

10 the experiments that do not focus on efficenty i find distracting they seem mere additional experiments illustrating prior work. If there is a result that the other two would not have produced you should compare

11 4.3 in paricular i find misleading u should byte level ensemble introduced by phan, so compare to them

**Questions:**

Please help me understand exaclty how your algorithm is different from phan, phan also uses a tree data structure, and i am understanding u than use token level sampling but i find the sections in the paper hard to follow. Hence i find it hard to verify its correctness.

So to summarize my main concerns

Clarity of writing needs improvement

Clarity of claim ideally much less claim but more precition

Experiments are too unfocussed on the problem you are trying to solve which to my understanding is the efficency part not the pbp

---

> ### Author Response · Authors · 2025-11-20
>
> We thank the reviewer for the thoughtful review and for recognizing that we present a “more efficient version of [an] exact solution to the PBP,” and that this efficiency is “crucial for its adoption [and] broad appeal.” We address each of your concerns and questions below and are eager to discuss further.
>
> ### 1. Clarification of efficiency improvement
>
> We improve the **inference-time efficiency** of exact algorithms for producing byte-level probability distributions from tokenizer-based LMs by 3x over the prior best method, Phan et al.
>
> ### 2. Clarification of Contributions
> We agree with the reviewer that we can make the contributions relative to prior exact methods more clear in the abstract and introduction. We plan to revise the paper by discussing prior approaches in the introduction and highlighting the efficiency improvement of ByteSampler as a significant contribution.
>
> To summarize our contribution: while exact solutions to the PBP have been proposed in the past, ByteSampler is **at least 3x more efficient than prior approaches.** In addition, **we support a larger set of models than Phan et al.** This makes ByteSampler the first computationally tractable (i.e. non-exponential time) exact solution to support byte-level BPE tokenizers.
>
> ### 3. Definition of exactness
> We define exactness precisely in Appendix D.3 as it is quite long and technical. Briefly, ByteSampler is exact under the assumption that the model does not generate invalid tokenizations. This definition is supported in prior work — indeed, it is the **same as Phan et al.’s**. We believe **this definition of exactness makes practical sense**, as it closely matches the standard practice of tokenizing the prompt (in the canonical fashion) and then sampling its completion in an unconstrained manner. Indeed, ByteSampler differs from normal sampling only in that it prevents the model from generating invalid token sequences, and we have no reason to believe that allowing models to commit tokenization errors during sampling is beneficial.
>
> ### 4. Comparison to Phan et. al.
> ByteSampler supports a different set of models from Phan et al., including the models that are popular today, and is also much more efficient.
>
> 1. ByteSampler supports a different set of models from Phan et al. Summarized in Table 1, Phan et. al. supports SentencePiece tokenizers (+ Gemma via a special case) while we support HuggingFace ByteLevel BPE tokenizers. We consider this an advantage of our solution because **we are able to support nearly all recently released models,** while to our knowledge, **no base model trained using a SentencePiece BPE tokenizer has been released since 2023.**
>
>     The reason for the differing model support boils down to a difference in algorithmic structure of our solution. The method of Phan et. al. is based on determining the last token in the sequence and then tokenizing the preceding text with the tokenizer. In contrast, **we build the tree forward, taking a stream of bytes and updating the tree state incrementally.** This allows us to adapt to tokenizers with complex structure, such as pretokenization that splits in the past based on the value of future bytes, a feature that is used by almost all modern models.
>
> 2. **ByteSampler is much more efficient than Phan et al.** Our method enables a 3.3 × reduction in inference tokens, due to the more aggressive pruning using the token validity constraints, while also reducing preprocessing complexity, due to the incremental construction of the covering tree. (Phan et al. [1] effectively rebuild the tree from scratch for every byte.)
>
> ### 5. Cover definition
> The algorithm of Phan et. al. does not use a tree structure and their cover is structured more like a collection of beams. We are using the notion of covering from Vieira et. al. (2024) [1] but augment the definition to only permit valid token sequences.
>
> [1] Vieira, Tim, et al. "From Language Models over Tokens to Language Models over Characters." *Forty-second International Conference on Machine Learning.*
>
> ### 6. Proposition 3.1
> This proposition was not shown by Phan et. al.. It was shown in a slightly different form in [2], but this source is not well-known and uses different notation, so we find it useful to include it here.
> [2] Hendrik van Antwerpen and Alexander Neubeck “So many tokens, so little time: Introducing a faster, more flexible byte-pair tokenizer”
>
> ### 9. Overhead definition
> “Overhead” measures how many additional tokens of inference are needed (compared to BPE) on average. (This is per byte scored.)
>
> ### 8. and 10. Purpose of non-efficiency experiments
> We run empirical experiments to demonstrate that our method performs correctly in practice, rather than being supported by theory alone.

---

> > ### Comment · Reviewer_KhEW · 2025-11-27
> >
> > (1) Claim: The authors algorithm supports more models.
> >
> > I am sorry I still do not understand that claim. The other algorithms support BPE, they may not have open sourced implementations of their work in Huggingface, but I find that irrelevant. There is no algorithmic difference.
> >
> >
> > (2) Can you update your paper with the changes you proposed, and mark the changes you have made. I would like to review the changes.
> >
> > I think the paper could meet the bar for publication, efficiency improvements alone are good enough for me. However, the over-claims, the consistency of argumentation and experimental evaluation worry me still. Hence my request. Let's discuss based on the updated version of the paper.

---

### Meta-Review · Area_Chair_FeDH · 2026-01-01

**Summary:**

This paper introduces the prompt boundary problem as a means of computing next-byte probabilities for autoregressive sampling, enabling the effective conversion of autoregressive language models with BPE tokenizers into character- or byte-level models. While the approach is interesting and potentially useful, the primary concern in the current submission is that the distinctions from prior work are not sufficiently clearly justified. Such problems were not solved well in the discussions and the revised version. For these reasons, I do not believe the paper is yet ready for publication in its present form. Nevertheless, I believe this work could develop into a strong and impactful contribution to the field when addressing these points.

**Reviewer Concerns:**

All reviewers have similar concerns about the contribution, novelty, and overlap with prior work. Such concerns are not well solved in the discussion, and no modifications are made in the revised version.

**Reviewer Scores:**

Reviewer KhEW: The score is kept at 4, as no revised version addressing the reviewer’s concerns was provided.

Reviewer Nkha: The score is increased to 4 but remains negative, as further effort is needed to clearly justify the distinctions from prior work.

Reviewer E5BB: The score is kept at 6, as key parts of the submission would benefit from further polishing, particularly in discussing related work and highlighting the paper’s unique contributions.

Reviewer Qz8v: The score is kept at 4, as the primary concern regarding overlap with previous work has not been sufficiently addressed.

---

### Decision · Program_Chairs · 2026-01-26

Reject